# Multicolored sequential resonance energy transfer for detection of simultaneous ligand binding at G protein-coupled receptors

Alice Valentini[1], Bethany Dibnah[2], Marija Ciba[1], Elaine M. Duncan[2], Asmita Manandhar[1], Bethany Strellis [2], Luca Vita[2], Olivia Lucianno[2], Conor Massey[2], Sophie Coe[2], Trond Ulven [1], Brian D. Hudson [2] ✉ & Elisabeth Rexen Ulven [1] ✉

G protein coupled-receptors (GPCRs) are the largest family of signalling proteins and highly successful drug targets. Most GPCR targeting drugs interact with a binding pocket for the natural ligand of the receptor, typically near the extracellular region of the transmembrane domains. Advancements in structural biology have identified additional allosteric binding sites in other parts of these receptors. Allosteric sites provide theoretical advantages, including the ability to modulate natural ligand function, and there is a need for better ways to study how ligands interact with these different GPCR binding sites. Here we have developed an approach to study multiple ligands binding to the same GPCR at the same time based on sequential resonance energy transfer between two fluorescent ligands. We use this approach to define ligand pharmacology and to understand binding kinetics at the FFA1 free fatty acid receptor, a clinically relevant receptor. This approach to study GPCR ligand interactions will aid development of new GPCR drugs.

G protein-coupled receptors (GPCRs) are the most successful drug targets, with estimates suggesting one third of drugs act through these receptors[1]. The success of GPCRs derives from their diverse biology and good understanding of how to develop drugs for these receptors. GPCRs share a conserved seven transmembrane structure, transmitting signals from extracellular ligands, including hormones, neurotransmitters and metabolites, into intracellular responses. Physiological GPCR ligands interact with the orthosteric binding site, commonly found at the top of the transmembrane helical bundle[2]. GPCR drug discovery has had great success identifying synthetic ligands that compete for binding at orthosteric sites, however recently, focus has shifted towards drugs binding to distinct allosteric sites[3].

Allosteric GPCR ligands provide conceptual advantages over orthosteric ligands. While orthosteric ligands primarily function as either agonists or antagonists, allosteric ligands may affect receptor

function in more diverse ways[4]. Some allosteric ligands function as agonists directly activating their receptor, but more commonly, allosteric ligands act by modulating the way physiological ligands bind or initiate signalling[5]. This means that allosteric modulators have a saturable effect, reducing the risk of adverse response, and will either enhance or supress function, while maintaining natural spatiotemporal control of signalling[6]. Attention has also been given to the possibility that allosteric ligands may bias signalling towards particular pathways[7], allowing for more fine-tuned control of signalling. While this diversity in allosteric ligand function creates opportunity for drug development, it also presents challenges to how we measure the effects these ligands have.

The revolution in GPCR structural biology led by advances in X-ray crystallography and cryogenic electron microscopy has defined the locations where many allosteric ligands bind to GPCRs[8]. These

[1]Department of Drug Design and Pharmacology, University of Copenhagen, Universitetsparken 2, Copenhagen DK-2100, Denmark. [2]Centre for Translational Pharmacology, School of Molecular Biosciences, College of Medical, Veterinary and Life Sciences, University of Glasgow, Glasgow G12 8QQ Scotland, United Kingdom. ✉e-mail: Brian.Hudson@Glasgow.ac.uk; eru@sund.ku.dk

locations are diverse, with examples of allosteric ligands binding to extracellular facing pockets as well as to extrahelical pockets facing the lipid bilayer. While structural approaches are important, they must be complimented with studies defining the ligand pharmacology. Considering allosteric ligands produce their effects in part by modulating the binding affinity of ligands binding to other sites, developing approaches able to measure and quantify simultaneous binding of multiple ligands to the same receptor would be of value.

Resonance energy transfer approaches including bioluminescence and Förster resonance energy transfer (BRET and FRET) are highly effective at measuring binding of fluorescently labelled ligands to GPCRs[9]. In particular, the development of NanoBRET, which measure energy transfer from a Nanoluciferase (Nluc) fused to the extracellular N terminal of the GPCR to a fluorescent ligand[10], has revolutionised GPCR ligand binding studies[11]. However, no method exists for probing the dynamic interplay between multiple ligands simultaneously interacting with a receptor. We propose that this could be achieved using multiple fluorescent ligands with overlapping spectral properties, allowing for sequential energy transfer from an Nluc to a fluorescent ligand binding at one site, then subsequent energy transfer to a second fluorescent ligand binding to a different site. Such an approach would allow for the direct measurement of ligand binding parameters including affinity and binding kinetics only from receptors with both fluorescent ligands simultaneously bound.

To demonstrate this approach, we characterise ligand binding to the free fatty acid receptor, FFA1, a clinically important GPCR with potential in the treatment of metabolic disorders[12]. The pharmacology of ligand binding to FFA1 is complex, with studies demonstrating at least two distinct sites on the receptor that each show cooperativity with fatty acid binding[13]. Structural studies established that one of these sites, often referred to as allosteric site one, resides near the traditional GPCR orthosteric site, but extends out of the helical core through a gap between TM3 and TM4[14]. Adding complexity, recent cryo-EM structures bound to fatty acid ligands suggest that this site is at least one site where fatty acids bind to the receptor[15]. The second binding site, often referred to as allosteric site two, is a lipid facing extrahelical pocket formed between the bottoms of TM3, TM4, TM5 and intracellular loop 2[16]. Ligands binding to site two are notable for having greater efficacy in G protein assay and therefore site one and site two ligands are sometimes referred to as partial and full FFA1 agonists, respectively[12]. Previous work has shown that NanoBRET can effectively measure ligand binding to FFA1 site one[17], while no fluorescent tracers have been reported for site two.

In this work we develop fluorescent tracers that bind to each of the two FFA1 binding sites. These tracers are designed to possess spectral properties to allow for sequential energy transfer[18]. We show that these ligands can be combined with a NanoBRET binding assay to measure their simultaneous binding to FFA1. This assay is then used to define ligand affinity, binding kinetics and pharmacology of the FFA1 receptor bound to two ligand simultaneously.

## Results

### Development of a green FFA1 site two tracer

To measure simultaneous ligand binding to FFA1, suitable tracers with affinity for both FFA1 binding sites are required. High affinity fluorescent tracers for site one are known[17,19,20], but no tracers have been described for site two. Given the structural similarities among ligands binding to the two FFA1 sites[13], establishing a screening strategy that confirms the likely binding site of developed tracers was important. Since site two ligands behave as full agonists in Gαq signalling, while site one ligands are partial agonists, a Ca²⁺ mobilisation assay was employed. Two reference site two compounds, T-3601386 (T360)[21] and Cpd 6h[22], show full agonism, while a reference site one compound, TUG-770[23], exhibited partial agonism (Supplementary Table 1). To

supplement this strategy, T360 and Cpd 6h, along with another reference site one agonist, TUG-905[24], were tested in an arrestin-3 recruitment assay (Supplementary Fig. 1A). In this assay the pharmacology was reversed, the site one compound was a full agonist, while the site two compounds were partial. To identify potential attachment sites and suitable linkers for fluorophore labelling, Cpd 6h served as template for the design. A series of analogues linked to a Boc-amine were synthesized and screened (Supplementary Table 1). This follows a strategy previously used, where the Boc-amine serves as a stable steric probe that, if tolerated, is deprotected and coupled to the desired fluorophore[25]. All compounds show full agonism in Ca²⁺ assays, consistent with binding to site two. Two compounds were identified with nanomolar potency and the best scope for conversion to fluorescent tracers, TUG-2457 and TUG-2467.

A green nitrobenzofurazan (NBD) fluorophore was incorporated into TUG-2457 and TUG-2467 to produce TUG-2489 and TUG-2490 (Fig. 1A). While both compounds appear to show full agonist activity in Ca²⁺ assays (Fig. 1B; Supplementary Table 1), their potencies are low (EC₅₀: 0.8-4 µM) making it difficult to fully determine efficacy. As TUG-2490 displayed higher potency, we aimed to improve this compound by replacing its phenyl substituent with a 6-methyl-2-pyridyl, as is present in Cpd 6h, resulting in TUG-2597 (Fig. 1A). Pleasingly, TUG-2597 has much-improved potency, shows full agonist activity in Ca²⁺ assays (Fig. 1B; Supplementary Table 1), and partial agonism for arrestin-3 recruitment (Supplementary Fig. 1B). Introduction of the heterocycle also improved aqueous solubility, which was a limiting factor with the previous analogues. To explore alternative sites for fluorophore addition, we also produced TUG-2598 (Fig. 1A), where the NBD is linked to the carboxylate of Cpd 6h, which according to the crystal structure does not participate in any ionic interactions[16]. This modification resulted in loss of activity (Fig. 1B), and therefore we proceeded with TUG-2597 as the best site two green tracer.

To use TUG-2597 in a NanoBRET-based binding assay, a version of FFA1 tagged with the Nanoluciferase (Nluc) BRET energy donor is required. Traditionally, NanoBRET GPCR binding assays employ receptors tagged at their N terminal, resulting in proximity to extracellular facing orthosteric sites[10]. However, binding site two of FFA1 resides within an extrahelical groove located near intracellular loop 2[16,26], and depending on where the fluorophore is attached and the nature and length of the linker, it is not obvious whether a tracer binding to this site would be closer to Nluc attached at the N or C terminal. The NBD of TUG-2597 is predicted to be situated in the middle of the membrane, sufficiently close for BRET to occur with either Nluc location (Fig. 1C). Functionality of the Nluc fusions was confirmed using site specific agonists, yielding comparable Ca²⁺ responses at untagged, N- and C- terminal Nluc FFA1 constructs (Supplementary Fig. 2A, B). After confirming that the function of TUG-2597 was also similar at all three FFA1 constructs (Fig. 1D), saturation NanoBRET experiments showed that TUG-2597 binding (Kd = 350−440 nM) could be measured using either Nluc location (Fig. 1E, F). Comparison of BRET signals suggested that more efficient energy transfer occurs using the N terminal location (Fig. 1G).

### Development of an FFA1 site one red tracer

Having identified a green site two tracer, we next aimed to identify a red tracer for site one. To be used in sequential energy transfer with TUG-2597, the tracer needs to both bind with high affinity and contain a fluorophore that could serve as FRET acceptor for the NBD in TUG-2597. We generated two putative red tracers based on a previously published FFA1 NBD containing tracer, TUG-1460 (Supplementary Table 2)[17], one with the red sulfo-cyanine5 (SulfoCy5), TUG-2355, and one with the orange-red emitting Cy3.5, TUG-2287 (Fig. 2A). When tested in Ca²⁺ assays, TUG-2355 retained activity but showed poor potency, while TUG-2287 displayed no measurable activity (Fig. 2B, Supplementary Table 2). To improve the SulfoCy5 tracer we designed a

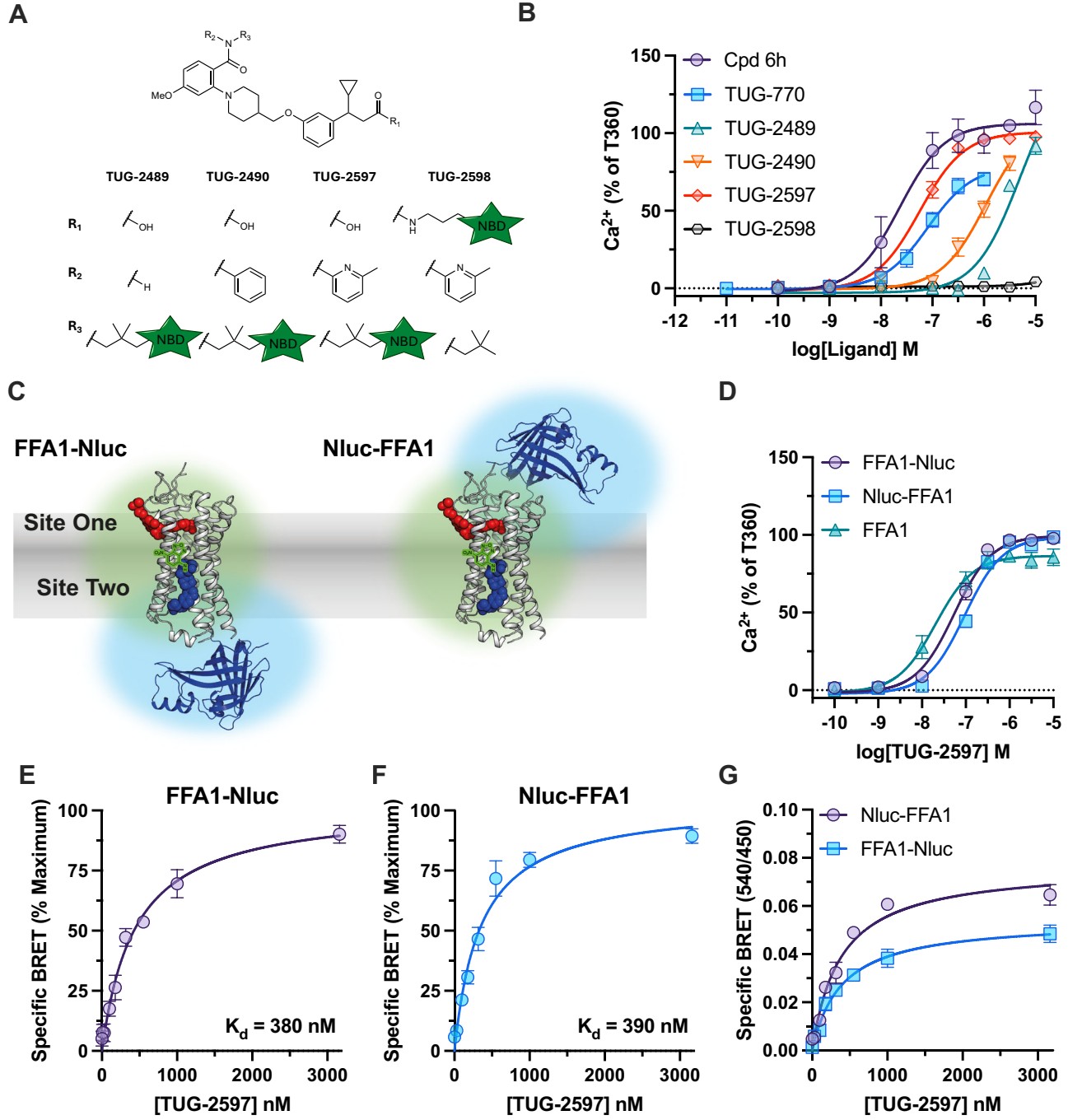

**Fig. 1 | Development of TUG-2597, a green fluorescent tracer binding to FFA1 site two. A.** Chemical structures of putative site two fluorescent tracers based on Cpd 6h. The linkers used and the position of an NBD green fluorophore are shown. **B** The activity of putative fluorescent tracers at FFA1 were assessed in $Ca^{2+}$ mobilisation assays and compared to FFA1 site two (Cpd 6h) and site one (TUG-770) reference ligands. Data are mean ± SEM from $N = 3$ independent experiments and normalized against the response to T360. **C** The proximity between the NBD tag of TUG-2597 (blue spheres) and Nluc is predicted to be sufficient for BRET to occur both with the Nluc tagged either at the C (left) or N (right) terminal of FFA1, shown with overlapping green and blue spheres in the illustration. A site one ligand is shown with red spheres. **D** Comparison of $Ca^{2+}$ responses to TUG-2597 in cells expressing FFA1 tagged at its N (Nluc-FFA1) or C (FFA1-Nluc) terminal with Nluc, to

cells expressing an untagged form of FFA1 (FFA1). Data are mean ± SEM from $N = 3$ (FFA1-Nluc, Nluc-FFA1) or $N = 4$ (FFA4) independent experiments and normalized against the response to T360. Specific BRET binding curve for TUG-2597 obtained in cells expressing an FFA1 construct tagged at its C (**E**) or N (**F**) terminal with Nluc are shown. Specific BRET was determined by subtracting non-specific BRET binding obtained in the presence of the competing ligand, T360 (10 µM). Data are expressed as a percentage of the maximal signal. **G** Saturation BRET binding curves for TUG-2597 are compared between the C terminal and N terminal FFA1 constructs. Data are shown as the BRET ratio (540 nm / 450 nm emission). Saturation binding experiments (**E–G**) show mean ± SEM from $N = 3$ independent experiments. Source data are provided as a Source Data file.

compound with shortened linker on either side of the amide connector, TUG-2591 (Fig. 2A), and were pleased to see improvement in potency (Fig. 2B, Supplementary Table 2). Like other site one agonists, TUG-2591 retained partial agonism in $Ca^{2+}$ assays relative to T360, and

full agonism in arrestin-3 assays (Supplementary Fig. 1C, D). TUG-2591 showed comparable activity across untagged, N and C terminal tagged FFA1 (Fig. 2C), while TUG-2355 and TUG-2287 have comparable activity at N and C terminal constructs (Fig. 2D, E).

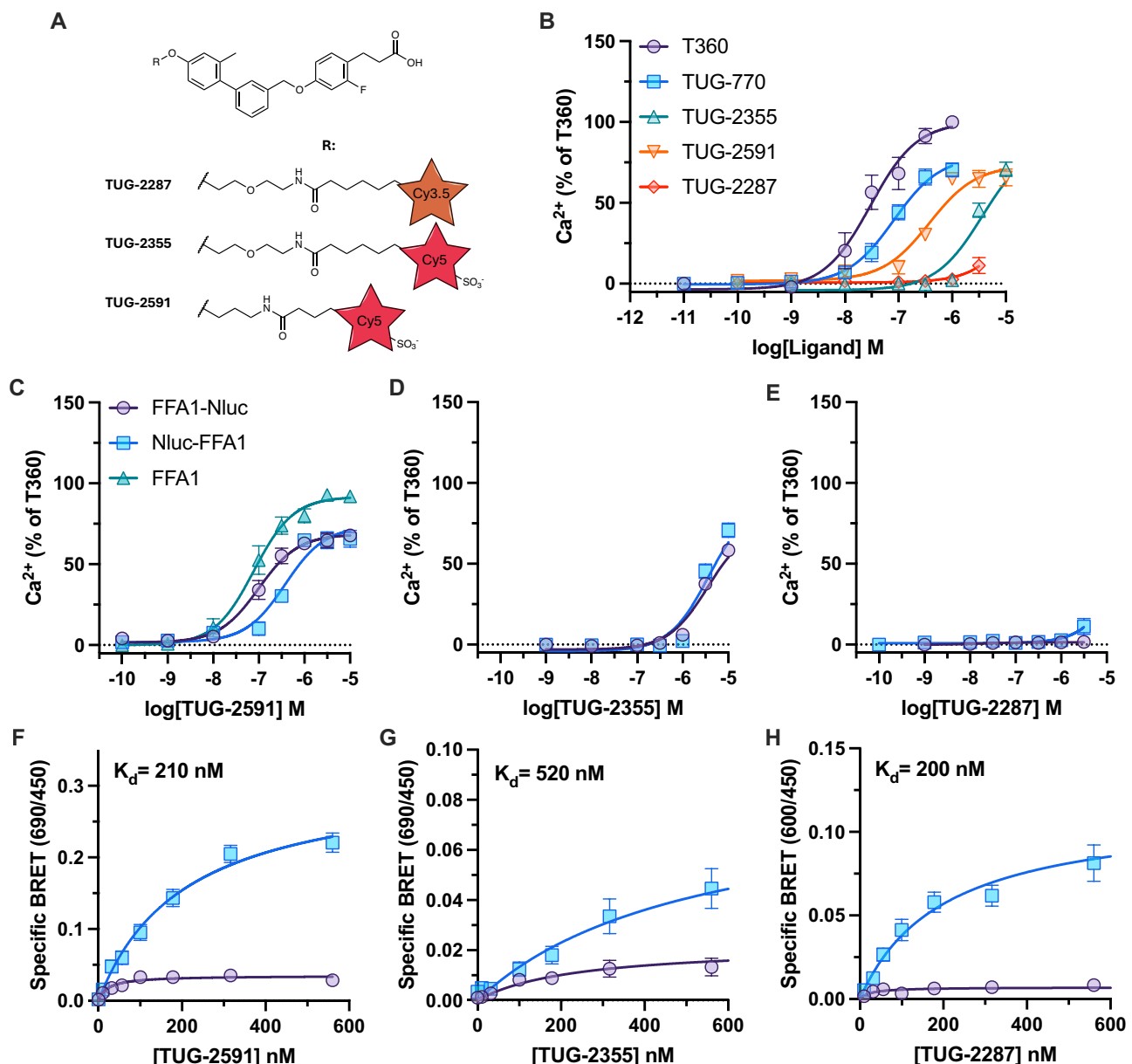

**Fig. 2 | Identification of TUG-2591 as a red fluorescent tracer binding to FFA1 site one. A** Chemical structures of putative red fluorescent tracers for site one of FFA1. Compounds are based on a previously described green tracer, TUG-1460, with the linkers used as well as the position and identity of the red fluorophore shown. **B** The activity of putative fluorescent tracers at FFA1 were assessed in $Ca^{2+}$ mobilisation assays and compared to T360 (site two) and TUG-770 (site one) reference ligands. Data are mean ± SEM from $N = 3$ independent experiments and normalized against the response to T360. The ability of the three putative red tracers, TUG-2591 (**C**) TUG-2355 (**D**) and TUG-2287 (**E**) to produce a $Ca^{2+}$ mobilisation responses are shown from cells expressing FFA1 tagged at its C (FFA1-Nluc) or N (Nluc-FFA1) terminal with Nluc or an untagged form of FFA1 (FFA1). Data in **C** are mean ± SEM from $N = 3$ (FFA1-Nluc, Nluc-FFA1) or $N = 4$ (FFA1), data in (**D**) are from $N = 3$, and data in (**E**) are from $N = 2$ independent experiments and normalized

against the response to T360. Saturation BRET binding experiments compare specific BRET obtained from the C (purple, circles) or N (blue, squares) terminal FFA1 Nluc constructs with TUG-2591 (**F**), TUG-2355 (**G**), and TUG-2287 (**H**). Specific BRET was calculated by subtracting non-specific BRET binding obtained in the presence of the competing ligand, TUG-770 (10 µM). Saturation binding curve fit parameters with 95% confidence intervals are: TUG-2591- FFA1-Nluc, $K_d = 23$ (15−35) nM, $B_{Max} = 0.035$ (0.032−0.038), Nluc-FFA1, $K_d = 210$ (150−300) nM, $B_{Max} = 0.31$ (0.27-0.37); TUG-2355- FFA1-Nluc, $K_d = 240$ (75−810) nM, $B_{Max} = 0.022$ (0.015−0.038), Nluc-FFA1, $K_d 5 = 20$ (260−1200) nM, $B_{Max} = 0.083$ (0.060−0.14); TUG-2287- FFA1-Nluc, $K_d = 25$ (1.2−130) nM, $B_{Max} = 0.007$ (0.005−0.01), Nluc-FFA1, $K_d = 200$ (120−340) nM, $B_{Max} = 0.11$ (0.95−0.14). Saturation binding experiments (**F−H**) are mean ± SEM from $N = 3$ independent experiments. Source data are provided as a Source Data file.

In NanoBRET binding assays using N terminal Nluc-FFA1, the results for the two SulfoCy5 compounds were as predicted, TUG-2591 displayed higher affinity than TUG-2355 (Fig. 2F, G). Unexpectedly, the Cy3.5 compound, TUG-2287, which was inactive in $Ca^{2+}$ assays displayed comparable affinity to TUG-2591 (Fig. 2H). These observations suggest the Cy3.5 fluorophore in TUG-2287 interferes with the intrinsic activity of the compound, resulting in a compound with affinity but no agonism. Importantly, across all three site one red tracers,

substantially less BRET was measured when using the C terminal Nluc construct compared to when using the same ligand at the N terminal Nluc construct (Fig. 2F−H).

## DISCo-BRET measures simultaneous ligand binding to distinct FFA1 sites

To measure simultaneous ligand binding to FFA1, a sequential resonance energy transfer approach has been employed. There is

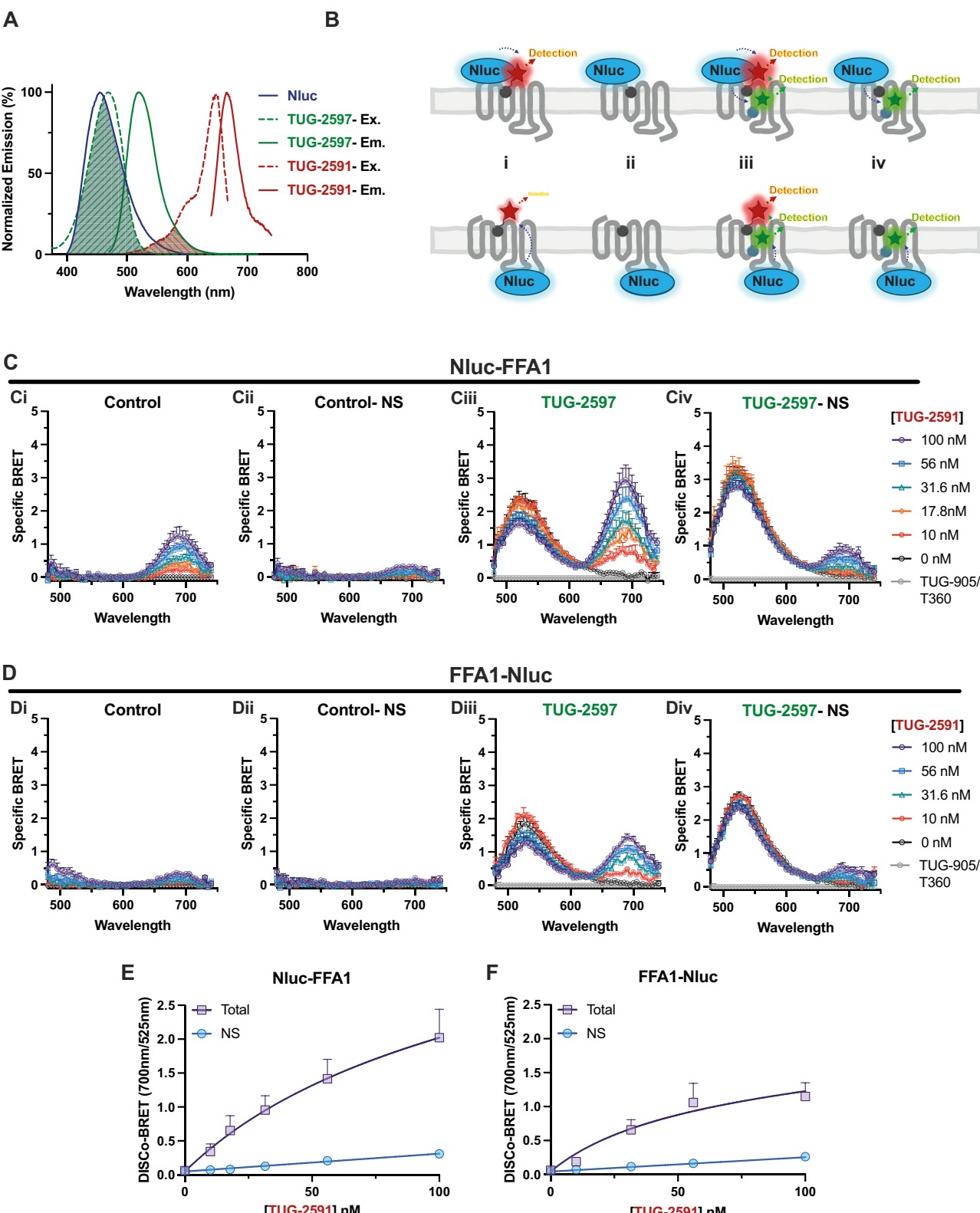

excellent spectral overlap between the emission of Nluc and the excitation of our green tracer, TUG-2597, as well as overlap between the emission of TUG-2597 and the excitation of our red tracer, TUG-2591 (Fig. 3A). This suggests sequential resonance energy transfer from Nluc to TUG-2597, then on to TUG-2591 may occur if both compounds bind FFA1 simultaneously. However, given the broad emission peak of Nluc, and potential for direct BRET between Nluc and TUG-2591, we hypothesised that increasing the distance between

Nluc and TUG-2591, which can be bridged by the NBD in TUG-2597, would contribute to energy acceptance from the green fluorophore rather than directly from Nluc. We have achieved this by using an FFA1 construct with Nluc at the intracellular C terminal of the receptor, and using a SulfoCy5 fluorophore in the red tracer, TUG-2591. Considering the hydrophilic nature of SulfoCy5 and the low level of membrane interactions of sulfonated cyanine dyes[27], we predicted that this should provide sufficient separation from the C

**Fig. 3 | Dual Interaction Sequential Compound (DISCo)-BRET measures simultaneous binding of tracer ligands to two distinct FFA1 binding sites.**
**A** The spectral properties of Nluc, TUG-2591 and TUG-2597 are shown. Spectral overlap in Nluc emission with TUG-2597 excitation, as well as overlap in TUG-2597 emission with TUG-2591 excitation are highlighted. **B** A cartoon diagram demonstrates predicted BRET energy transfer between Nluc, a red site one tracer and a green site two tracer. The cartoons depict the experimental conditions used incubating Nluc-FFA1 or FFA1-Nluc cells with: i) the red site one tracer alone; ii) the red site one tracer combined with a competing ligand; iii) the red site one tracer and the green site two tracer; and iv) the red site one tracer, the green site two tracer and a competing ligand for site one. **C** Luminescent emission spectra were recorded from cells expressing an N terminally Nluc tagged FFA1 with increasing concentrations of the red tracer TUG-2591. All spectra are shown as 'specific BRET' by subtracting the spectrum obtained in the presence competing ligands for both site one and two, TUG-905 (10 µM) and T360 (10 µM). Data are shown for the control condition (**Ci**); for the control non-specific (NS) condition, in the presence of TUG-770 to show non-specific TUG-2591 binding (**Cii**); in the presence of 316 nM of the green tracer TUG-2597 (**Ciii**); or non-specific binding in the presence of TUG-2597, with both 316 nM TUG-2597 and 10 µM TUG-905 (**Civ**). **D** Comparable experiments to those presented in **C** were conducted using cells expressing an FFA1 construct with Nluc fused to the C terminal of the receptor. DISCo-BRET saturation binding curves showing total and non-specific (in the presence of 10 µM TUG-905) TUG-2591 binding in the presence of TUG-2597 at either N (**E**) or C (**F**) terminal Nluc tagged FFA1 constructs are shown. DISCo-BRET was calculated by taking the area under the SulfoCy5 peaks (660-730 nm) divided by the area under the NBD peaks (490-560 nm) from the specific BRET spectra shown in (**C** and **D**). All DISCo-BRET data shown are mean ± SEM from $N = 4$ independent experiments. Source data are provided as a Source Data file.

terminal Nluc. Given that TUG-2597 binds to an extrahelical site within the bilayer, we hypothesised that this should provide the optimal condition for multi-coloured Dual Interaction Sequential Compounds (DISCo)-BRET, measuring energy transfer from the intracellular Nluc, through green TUG-2597 and on to red TUG-2591 (Fig. 3B).

To measure DISCo-BRET, cells expressing FFA1 tagged at its N (Fig. 3C) or C (Fig. 3D) terminal with Nluc were incubated in increasing concentrations of the red site one tracer, in the absence or presence of a near $K_d$ concentration of the green site two tracer. Luminescent emission spectra were obtained, confirming that in the absence of green tracer, specific BRET emission from the red tracer was observed at 690 nm when using the N terminal construct (Fig. 3Ci-Cii). In contrast, little BRET was observed with the C terminal construct (Fig. 3Di-Dii). Addition of the green tracer, resulted in a clear increase in BRET at 540 nm using either the N (Fig. 3Ciii) or C (Fig. 3Diii) terminal construct, indicative of direct BRET from Nluc to green tracer. Critically, addition of the green tracer also resulted in a clear increase in emission at 690 nm, again when using either the N (Fig. 3Ciii-Civ) or C (Fig. 3Diii-Div) terminal Nluc construct. To quantify this sequential energy transfer, we measured the area under the two emissions peaks, plotting the ratio as a saturation binding curve for the red tracer, TUG-2591 (Fig. 3E, F), demonstrating specific DISCo-BRET can be measured using either construct, with $K_d$ values for TUG-2591 of 50–100 nM. To simplify the experiment, we also measured DISCo-BRET by discretely measuring emission at three wavelengths, corresponding to Nluc, NBD and SulfoCy5 and plotting ratios of these emissions (Supplementary Fig. 3). These data demonstrate that while DISCo-BRET can be measured using simple BRET ratios, BRET between Nluc and TUG-2591 when using the Nluc-FFA1 construct makes it difficult to distinguish green tracer dependent DISCo-BRET from green tracer independent direct BRET (Supplementary Fig. 3A). In contrast, green tracer dependent DISCo-BRET was clearly observed using the FFA1-Nluc construct (Supplementary Fig. 3B). To extend these studies and explore how DISCo-BRET is affected by the concentration of green tracer, these experiments were repeated using a near saturating concentration of TUG-2597 (Supplementary Fig. 3C). These experiments show DISCo-BRET measures comparable affinity for TUG-2591 regardless of the concentration of TUG-2597 used. Although we found DISCo-BRET could be measured using either the ratio of SulfoCy5/Nluc (Supplementary Fig. 3Bi and Ci) or SulfoCy5/NBD (Supplementary Fig. 3Biii and Ciii) emission wavelengths, we have chosen to use the SulfoCy5/NBD ratio for further experiments.

## DISCo-BRET differentiates FFA1 ligands based on binding site and function

To demonstrate the potential of DISCo-BRET to characterise FFA1 ligands, a selection of previously reported compounds was tested in competition binding studies utilising near $K_d$ concentrations of each

tracer. These competing ligands included a mixture of site one agonists: TUG-905[24], TUG-770[23], TAK-875[28], and Merck Cpd-B[29]; one long chain fatty acid, α-linolenic acid (aLA); and two site two compounds, T360[21] and Cpd 6h[22]. We also tested an antagonist with unclear binding site, GW1100[30], a presumed orthosteric antagonist, PPTQ[31], and several analogues of PPTQ (Supplementary Fig. 4). DISCo-BRET binding studies were carried out with both the red and the green tracers along with increasing concentrations of the competing ligand. DISCo-BRET was measured using SulfoCy5/NBD (Fig. 4A), while binding of the green tracer was monitored using NBD/Nluc (Fig. 4B). BRET spectra were also obtained for each competing ligand to visualise DISCo-BRET (Supplementary Fig. 5). All site one agonists resulted in complete competition of DISCo-BRET, indicative of competitive binding with the red tracer (Fig. 4Ai). These site one agonists also enhanced BRET with the site two green tracer, consistent with binding cooperativity, a conformational change resulting in enhanced BRET between the C terminal Nluc and the green tracer, or increased green tracer emission due to a loss of sequential energy transfer to the red tracer (Fig. 4Bi). Clear differences in affinity across these compounds were observed, where TUG-905 > TAK-875 > TUG-770 > Cpd-B (Supplementary Table 3). The fatty acid ligand, aLA, showed little DISCo-BRET competition (Fig. 4Ai), and almost no effect on green tracer binding (Fig. 4Bi), likely due to low affinity for fatty acids at either site. In contrast, the site two compounds, T360 and Cpd 6h, display clear competition in both DISCo-BRET and green tracer binding (Fig. 4Aii and 4Bii). It is notable that these compounds did not fully compete in DISCo-BRET (Fig. 4. Aii; Supplementary Table 3), likely due to a low-level of direct BRET from the C terminal Nluc to the red tracer. The fact that T360 and Cpd 6h show competition in DISCo-BRET can only be explained by the ability of these ligands to displace the green tracer and therefore disrupt the sequential energy transfer to TUG-2591, and the residual BRET indicates that T360 and Cpd 6h are binding to site two rather than directly competing with TUG-2591.

When testing the antagonists, DISCo-BRET competition experiments demonstrated diverse pharmacology (Fig. 4Aiii and Biii). GW1100 showed no competition or enhancement for either DISCo-BRET or green tracer binding, suggesting that this antagonist binds a site distinct from site one or two. Despite this, GW1100 did functionally antagonise both site one and site two agonists (Supplementary Fig. 6A, B). In contrast, PPTQ and one of its derivatives, TUG-2743, appeared to show competition with both tracers (Fig. 4Aiii and Biii), supported by the fact that PPTQ antagonised both site one and site two agonists (Supplementary Fig. 6C, D). Although possible that PPTQ could bind to both sites, Schild antagonism experiments with PPTQ suggest its inhibition is only competitive with site one (Supplementary Fig. 6E, F), suggesting PPTQ is a site one ligand with negative allosteric modulation of site two. More interestingly, analogues of PPTQ with extended aromatic core, TUG-2744 and TUG-2745, displayed unique pharmacology, where they showed competition in DISCo-BRET, but had no effect on the green tracer (Fig. 4Aiii and Biii, Supplementary Table 3). This was supported

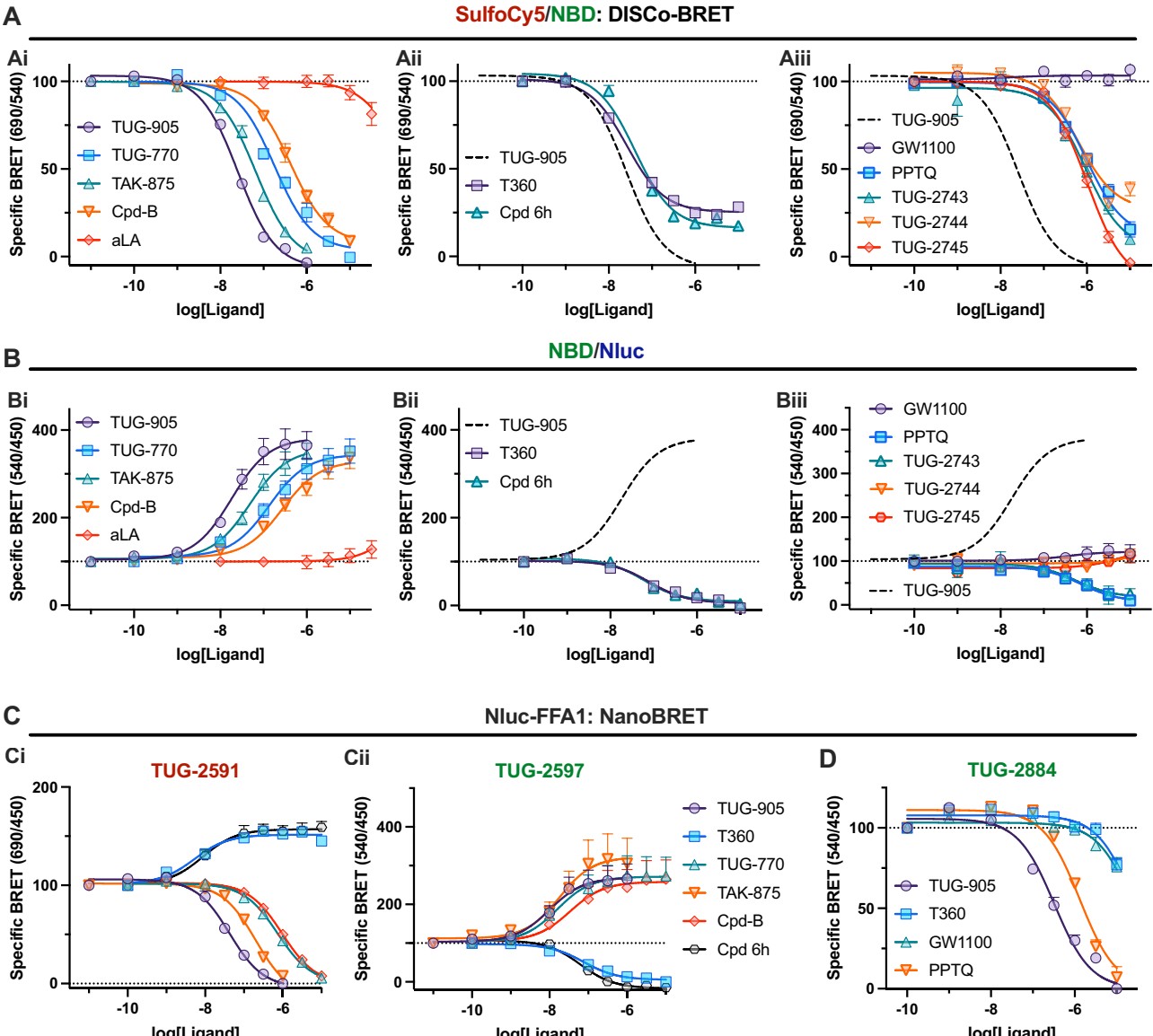

**Fig. 4 | DISCo-BRET competition binding experiments identify ligands binding to the two distinct FFA1 binding sites.** Competition DISCo-BRET binding experiments were conducted in the presence of both the red tracer TUG-2591 (100 nM) and the green tracer TUG-2597 (316 nM). Data are shown as the ratio of 690 nm / 540 nm luminescent emission (**A**), or 540 nm / 450 nm luminescent emission (**B**) and expressed as specific BRET above that obtained in the presence of both competing ligands, TUG-905 (10 μM) and T360 (10 μM). Competition binding data with various compounds are split over three graphs (**i - iii**), with the curve for TUG-905 shown as a dashed line in **ii** and **iii** for reference. All competition binding data are fit to a one site competition binding model, however given the complexity

of allosteric interactions between ligands binding to site one and two, it is recognised that this model does not fully describe the binding reality. DISCo-BRET competition data are mean ± SEM from N = 10 (TUG-905), N = 7 (TUG-770 and T360), N = 5 (Cpd 6h), or N = 4 (all other compounds) independent experiments. NanoBRET competition binding experiments using the N terminal Nluc tagged FFA1 construct and either 100 nM TUG-2591 (**Ci**) or 316 nM TUG-2597 (**Cii**), or 2 μM TUG-2884 (**D**) are shown. Specific binding is shown after subtracting BRET obtained with 10 μM TUG-905 (**Ci and D**), or 10 μM T360 (**Cii**). NanoBRET data are mean ± SEM from N = 3 (**Ci**), N = 4 (**Cii**), or N = 7 (**D**) independent experiments. Source data are provided as a Source Data file.

by antagonism studies, confirming that TUG-2745 inhibits signalling from site one, but not site two agonists (Supplementary Fig. 6G, H). These data suggest that extension of the antagonist core results in a loss of allosteric modulation of site two and demonstrates how DISCo-BRET allows a single competition binding assay to reveal complex pharmacology.

To support our observations in DISCo-BRET experiments, separate NanoBRET competition assays using Nluc-FFA1 were used to show that both T360 and Cpd 6h actually have positive cooperativity with the red tracer (Fig. 4Ci; Supplementary Table 3). As expected, NanoBRET binding assays for the green tracer (Fig. 4Cii; Supplementary Table 3) show broadly the same pattern of competition/

enhancement as observed for this tracer when measuring the NBD/Nluc ratio in the dual tracer DISCo-BRET assay (Fig. 4B; Supplementary Table 3), indicating that the enhancement observed in NBD/Nluc ratio in dual tracer DISCo-BRET experiments when adding unlabelled site one ligands (Fig. 4Bi), is not primarily due to reduced sequential energy transfer. To support our data with FFA1 antagonists, an FFA1 NBD antagonist tracer, TUG-2884, was generated based on TUG-2744 (Supplementary Fig. 7A) and shown to bind to Nluc-FFA1 (Supplementary Fig. 7B). Competition NanoBRET studies with TUG-2884 clearly establish that this compound is competed by PPTQ and a site one agonist, but not by GW1100 or a site two agonist (Fig. 4D, Supplementary Table 3).

## Allosteric regulation of FFA1 ligand dissociation

To assess how allosteric interactions affect binding kinetics, we explored dissociation binding kinetics using DISCo-BRET. Both tracers were co-added, followed by the addition of high concentrations of competing ligands for site one or site two to stimulate dissociation from one, the other, or both sites (Fig. 5A). Tracer binding was monitored as either SulfoCy5/NBD DISCo-BRET (Fig. 5B), or as NBD/Nluc to measure binding of the green tracer (Fig. 5C). Comparison of the DISCo-BRET dissociation curves obtained with different displacing ligands allowed for the determination of off-rate kinetics of each ligand from receptors bound to both tracers simultaneously (Fig. 5D and Supplementary Table 4). It was notable that the off-rate for the red TUG-2591 (from SulfoCy5/NBD when displacing with TUG-905) was substantially slower than that of the green TUG-2597 (from NBD/Nluc when displacing with T360) (Supplementary Table 4). Comparing the overall dissociation rate for our green tracer (Fig. 5E), obtained using the NBD/Nluc ratio (accounting for FFA1-Nluc co-bound with red TUG-2591 or not), suggested that the dissociation rate for the green tracer is broadly the same as when measured only from receptors co-bound to the red tracer (Supplementary Table 4). Because these experiments involved long agonist treatments, a bystander BRET internalisation assay was used to show that none of the ligands led to significant internalisation of FFA1-Nluc over the experimental time-course (Supplementary Fig. 8). To confirm that TUG-2591 does not affect TUG-2597 dissociation, association/dissociation experiments for TUG-2597 were carried out in single ligand NanoBRET format (Fig. 5F), which demonstrated a similar off-rate (Supplementary Table 4). While these data suggest the dissociation of our green site two tracer is not greatly affected by red tracer binding, the slow DISCo-BRET dissociation rate led us to also examine the binding kinetics of our red tracer with single ligand NanoBRET (Fig. 5G). Surprisingly, these experiments demonstrate that the off-rate of our red tracer is ~10 times faster for the single ligand experiment compared with dual ligand DISCo-BRET experiments (Supplementary Table 4).

To extend these studies and establish how the proportion of receptors bound to tracer impacts the way DISCo-BRET measures allosterism, we repeated these association-dissociation studies using near saturating concentrations of both tracers (Supplementary Fig. 9A, B). When assessing dissociations rates, we observe that the measured rates following addition of each competing ligand are comparable regardless of concentration of tracer ligands used (Supplementary Fig. 9C, D, Supplementary Table 4). This makes sense, as unlike a traditional allosterism experiment where saturating concentrations of the modulator are required, DISCo-BRET only measures from receptors with both ligands bound, therefore allosteric effects will be apparent at sub saturating modulator concentrations. However, we do observe that the rate of increased NBD/Nluc BRET following addition of the site one competing ligand, TUG-905, is much slower (Supplementary Fig. 9E). This is presumably because at near saturating concentrations of site one tracer, this tracer must dissociate before TUG-905 can bind. The fact that TUG-905 still enhances the NBD/Nluc BRET when using near saturating concentrations of green site two tracer, suggests that a component of this increase is due to conformational changes in the receptor, instead of modulation of affinity. To explore interaction between the two sites further, we conducted NanoBRET kinetic binding experiments assessing binding of the site two green tracer, TUG-2597 at either N or C terminal Nluc tagged FFA1 (Supplementary Fig. 9F, G). After the ligand had associated, a saturating concentration of site one ligand, TUG-905, was added, resulting in an immediate increase in BRET. This is in clear contrast to the equivalent DISCo-BRET experiment (Supplementary Fig. 9B), where this increase occurred over a prolonged period. Comparable experiments, suggest that the addition of T360 binding to site two, did not lead to a clear change in red, TUG-2591, ligand BRET when using a near saturating concentration of this ligand (Supplementary Fig. 9H). However, both TUG-2597

and TUG-2591 show clear slowing of dissociation rate when saturating concentrations of TUG-905 and T360 are present respectively (Sup Fig. 9I–K, Sup Table 4), demonstrating positive cooperativity between the unlabelled ligands and the tracers. These findings demonstrate that DISCo-BRET can define how ligand binding to one site of a GPCR affects dissociation at a second site, and importantly that it can do so using sub saturating concentrations of ligand.

## Allosteric regulation of FFA1 ligand association

Having demonstrated the potential for DISCo-BRET to study dissociation kinetics, we next explored binding association. Cells were first incubated with red tracer, TUG-2591, followed by addition of the green tracer, TUG-2597 (Fig. 6A). Binding was assessed either using the DISCo-BRET SulfoCy5/NBD ratio (Fig. 6B), or the NBD/Nluc ratio measuring binding of the green tracer (Fig. 6C), both in the absence and presence of site one competing ligand TUG-905. As expected, little increase in DISCo-BRET was seen following incubation with red tracer, while a rapid increase was observed with addition of green tracer (Fig. 6B). All DISCo-BRET was blocked by TUG-905. When assessing site two binding (Fig. 6C), association was observed after addition of TUG-2597, the signal for which was enhanced by the presence of TUG-905. Analysis of the DISCo-BRET association rate observed upon addition of TUG-2597 compared with the rate observed from NBD/Nluc, and with the association rate of TUG-2597 from a single ligand NanoBRET experiment, indicated that all three rates were identical (Supplementary Fig. 10A; Supplementary Table 5). This indicates DISCo-BRET association observed is due to TUG-2597 binding at site two to receptors that were pre-bound to the site one tracer. The fact that the single ligand experiment also shows a similar association rate, indicates that association rate of TUG-2597 is not affected by TUG-2591. This was further confirmed by single ligand NanoBRET association studies using multiple concentrations of green tracer (Supplementary Fig. 10B, Supplementary Table 5).

Finally, to examine how binding of green TUG-2597 to site two affects binding of red TUG-2591 to site one, experiments were conducted where the green tracer was added first, followed by the addition of the red tracer (Fig. 6D). As before, binding of green tracer was measured using the NBD/Nluc BRET ratio (Fig. 6E), while binding of the red tracer was assessed through the DISCo-BRET SulfoCy5/NBD ratio and experiments were conducted in the absence or presence of site one competing ligand, TUG-905 (Fig. 6F). These results demonstrated a rapid increase in NBD/Nluc green tracer signal, which was not impacted by the addition of red tracer, although green tracer signal was enhanced by pre-treatment with TUG-905 (Fig. 6E). While it may have been expected that the NBD/Nluc ratio would decrease due to sequential energy transfer to the red tracer, this likely is not observed because these experiments have been carried out at subsaturating concentrations of the two tracer ligands. However, addition of red tracer did lead to an increase in DISCo-BRET, with a relatively slow association rate (Fig. 6F; Supp. Table 5). To define the association rate of our red tracer at the pool of FFA1 receptors pre-bound to green TUG-2597, DISCo-BRET association experiments were conducted pre-treating with the green tracer, followed by addition of multiple concentrations of red tracer (Fig. 6G, Supplementary Fig. 11). Isolation of the TUG-2591 association kinetics allowed for calculation of binding parameters broadly in line with those calculated from the single concentration experiment (Fig. 6H; Supplementary Table 5). However, when comparable NanoBRET binding was used to assess association kinetics of TUG-2591 in the absence of TUG-2597, substantially faster association rates were observed (Fig. 6I; Supplementary Table 5). Indeed, the association rate of TUG-2591 was approximately ten times slower when measured using DISCo-BRET to assess binding to receptors prebound with TUG-2597, than it was when binding to a ligand free receptor. Interestingly, this broadly corresponds to the reduction in off rate seen for TUG-2591 when TUG-2597 is bound, and as a result the

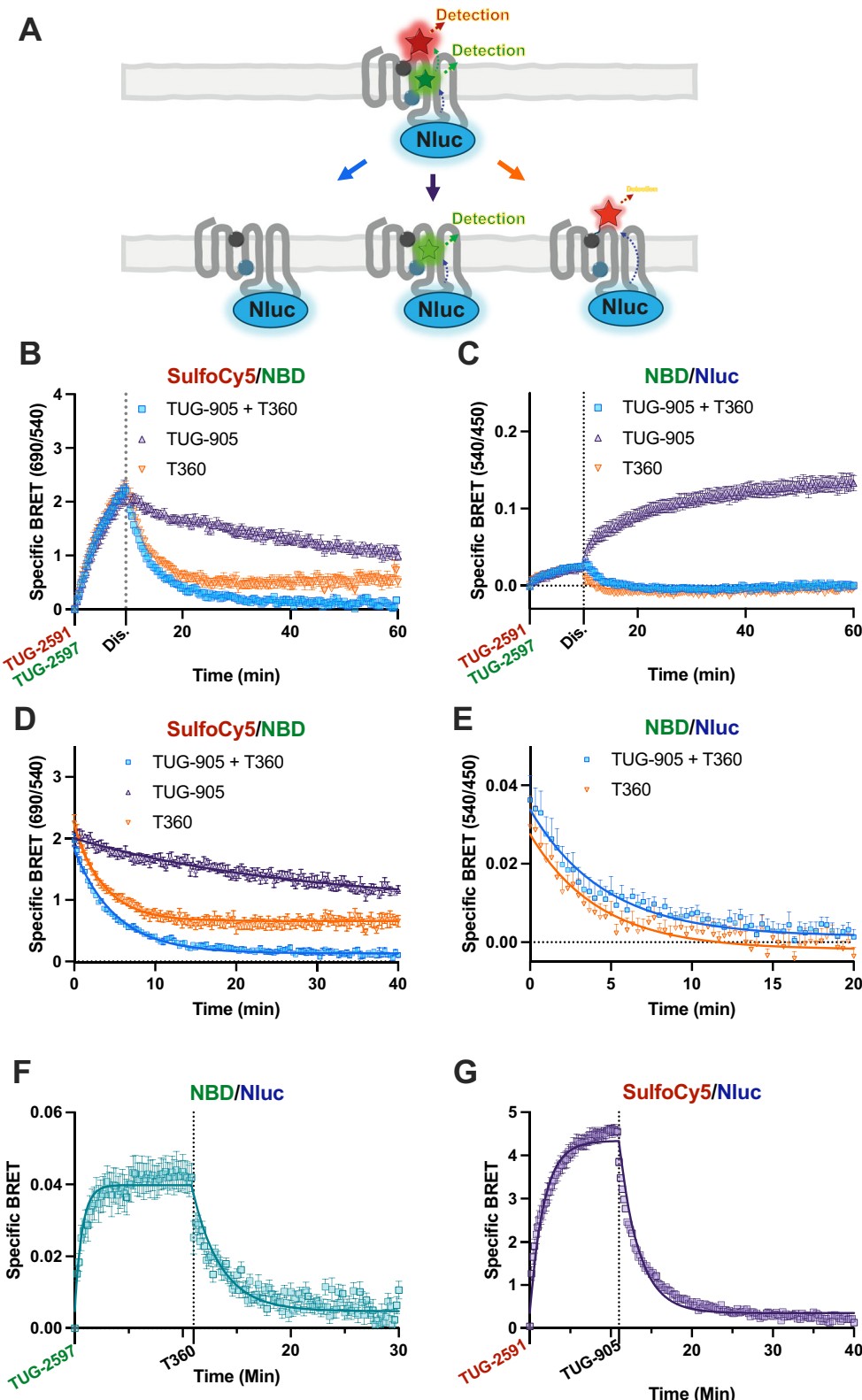

overall affinity of TUG-2591 for site one remains largely unchanged regardless of whether green TUG-2597 is bound to site two or not (Supplementary Table 5).

## Discussion

GPCRs are the largest and most important family of drug targets. In recent times focus on these receptors has shifted towards ligands that interact with allosteric sites, distinct from the binding site of the physiological ligand for the receptor[5]. We have developed a multi-coloured BRET approach that allows for simultaneous measurement of ligand interaction at multiple GPCR binding sites. We show this approach can characterize complex ligand pharmacology at the FFA1 receptor and define how ligands interact with each other to affect affinity and kinetics.

BRET as a technology has become an invaluable tool in GPCR research and BRET biosensors can measure nearly every aspect of

**Fig. 5 | DISCo-BRET kinetics reveals a slow off rate at site one when a compound is bound to site two. A** A cartoon diagram depicting a DISCo-BRET dissociation-rate experiment. Receptor was pre bound to both the red and green tracers, followed by addition of a competing ligands for both site one and site two (left), site one alone (middle), or site two alone (right). Experiments were conducted where cells were first treated with TUG-2591 (100 nM) and TUG-2597 (316 nM) together, before the addition (marked as Dis.) of TUG-905 (10 μM), T360 (10 μM) or a combination of TUG-905 and T360 (10 μM each) to assess the dissociation of the FFA1/ TUG-2591/ TUG-2597 complex. Data from these experiments are shown

as the 690 nm / 540 nm (**B**), or 540 nm / 450 nm (**C**) luminescent emission ratios. **D** The dissociation portion of the curve from (**B**) is plotted and fit to a one site exponential decay model. **E** The dissociation portion of the curve from (**C**) is plotted and fit to a one site exponential decay model. Data in (**B**–**E**) are from $N = 3$ independent experiments. Dissociation NanoBRET experiments are shown for TUG-2597 (**F**) and TUG-2591 (**G**), following treatment with T360 (10 μM) and TUG-905 (10 μM) respectively. NanoBRET data are mean ± SEM from from $N = 3$ (**F**) or $N = 4$ (**G**) independent experiments. Source data are provided as a Source Data file.

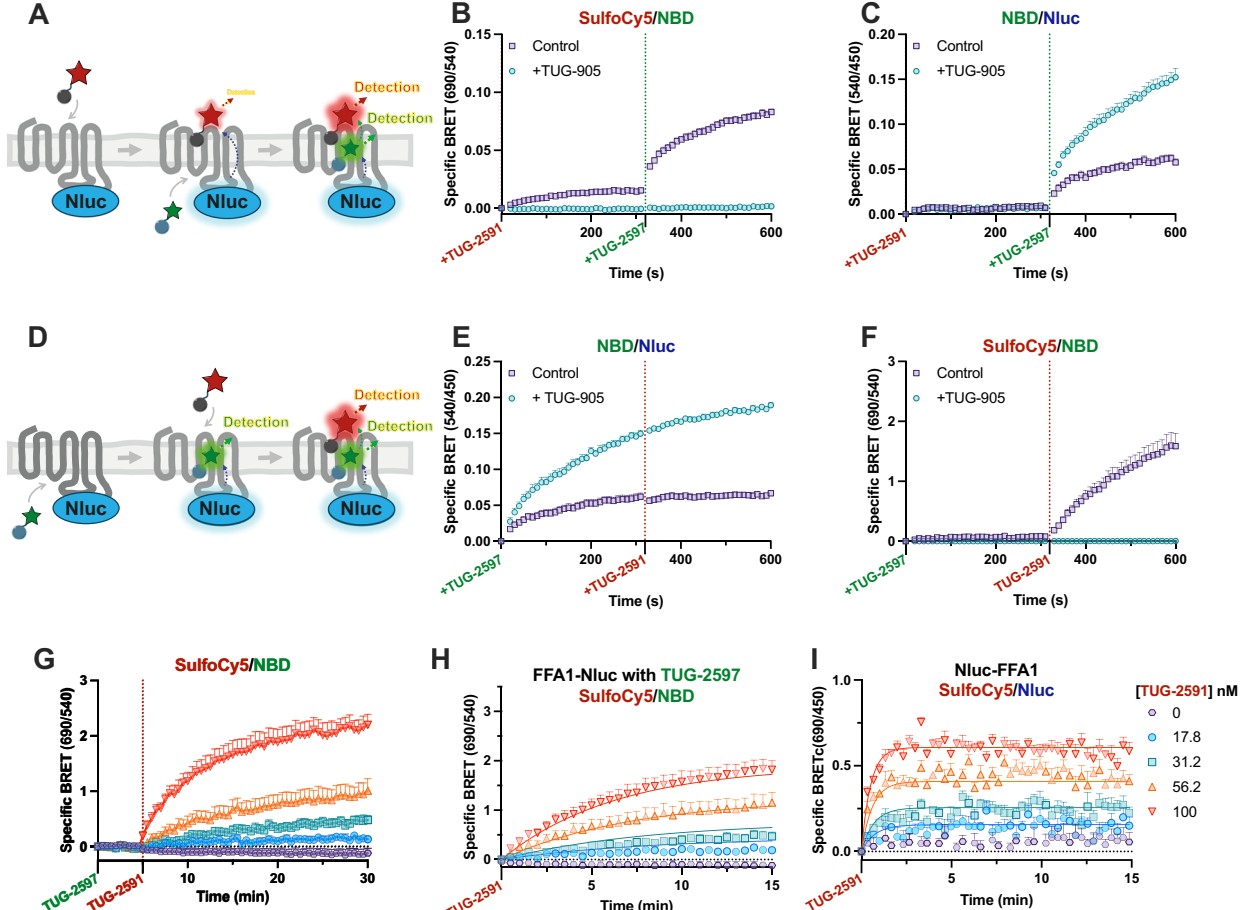

**Fig. 6 | DISCo-BRET demonstrates significantly slower on-rate kinetics at FFA1 site one when a compound is bound to site two.** Sequential binding experiments were conducted by treating first with red TUG-2591 (100 nM), then with green TUG-2597 (316 nM) (**A**), and data were plotted as either the 690 nm / 540 nm (**B**) or the 540 nm / 450 nm (**C**) luminescent emission ratios. Experiments were carried out either as control experiments or in the presence of TUG-905 (10 μM) to prevent binding of TUG-2591. Specific BRET was obtained by subtracting the non-specific BRET ratio obtained in cells that had been pre-treated with both TUG-905 (10 μM) and T360 (10 μM). Data in (**B**, **C**) are mean ± SEM from $N = 4$ (Control) or $N = 3$ (TUG-905) independent experiments. Comparable sequential binding experiments were conducted treating first with TUG-2597 (316 nM), then with TUG-2591 (100 nM) (**D**

–**F**). Data in (**E**, **F**) are mean ± SEM from $N = 5$ (Control) or $N = 3$ (TUG-905) independent experiments. **G** Sequential binding experiments adding first TUG-2597 (316 nM) followed by increasing concentrations of TUG-2591. Data are mean ± SEM from $N = 3$ independent experiments. **H** The TUG-2591 association portion of these experiments are isolated and fit to a multiple concentration of labelled ligand binding association model. Data are mean ± SEM from $N = 3$ independent experiments. **I** Comparable NanoBRET association experiments using an N terminal Nluc tagged FFA1 construct and multiple concentrations of TUG-2591 are fit to a multiple concentration of labelled ligand binding association model. Data are mean ± SEM from $N = 4$ independent experiments. Source data are provided as a Source Data file.

GPCR function[32]. NanoBRET based binding assays have in particular provided opportunity for improved GPCR ligand binding studies[10]. Advancements of the NanoBRET technique have allowed for assessment of ligand binding to GPCRs in vivo[33], and to receptors that are expressed through endogenous promoters[34]. The approach has also facilitated studies examining ligand binding to receptor oligomers[35], or measuring binding to intracellular binding sites[36–38]. Efforts have also started to explore how different ligands binding to the same site can be

multiplexed, through the use of different fluorophores[39]. However, these studies do not allow for simultaneous detection of multiple ligands binding to the same receptor. To achieve this, we have employed an innovative multi-colour sequential BRET strategy that we name DISCo-BRET. Historically, multi-colour BRET studies have employed multiplex strategies, where a single luciferase separately donates energy to two different acceptors[40,41]. While this is useful, it does not require both acceptors to be present for final acceptor

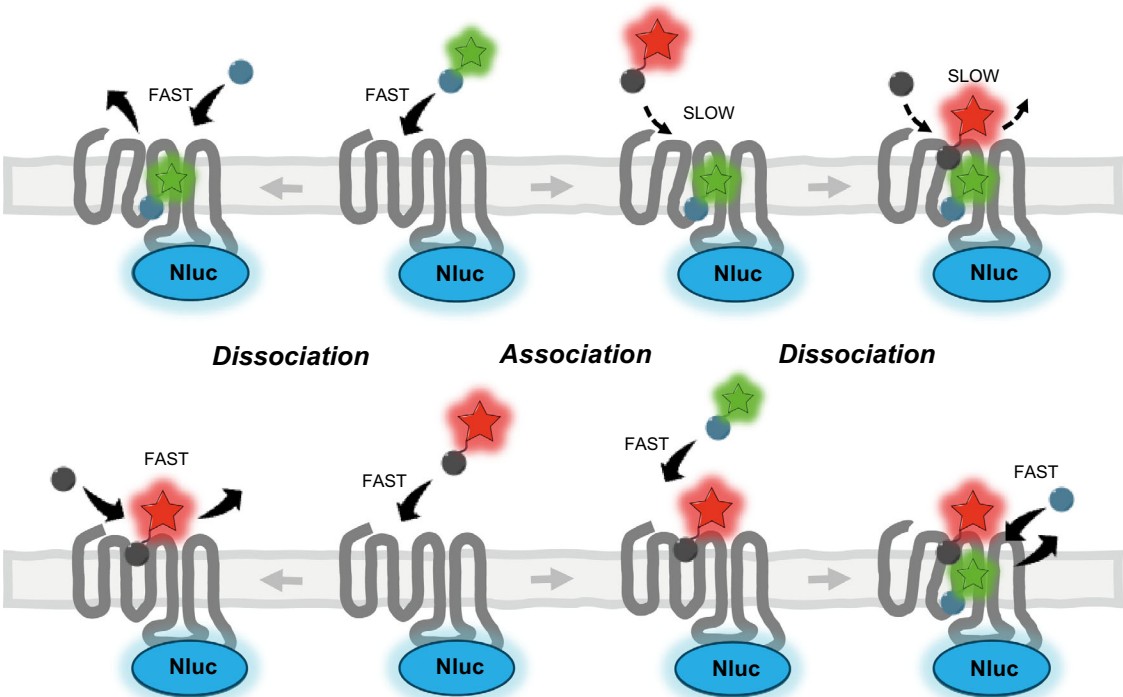

**Fig. 7 | Cartoon summarising site one and site two tracer binding kinetics.** Top shows fast binding of green site two tracer leads to a conformational change in the receptor. This likely close off access to the site one red tracer leading to both slow association and dissociation. Bottom shows fast association of the red site one tracer does not result in any change to the binding site for the green tracer, maintaining both fast association and dissociation.

emission, and therefore does not allow measurement only when the donor and both acceptors are all in close proximity. Employing sequential resonance energy transfer, where energy passes from donor to a first acceptor, then on to the final acceptor addresses this issue. However, very few studies have successfully employed sequential energy transfer BRET, even when measuring protein-protein interactions, with the best example having used the approach to demonstrate a heterotrimer among three GPCRs[42].

The biggest challenge to sequential resonance energy transfer is the wide emission band of luciferase enzymes, making it difficult to identify a final energy acceptor that does not directly accept energy from the luciferase. While a natural approach to address this might be to employ longer wavelength fluorophores, this is often limited by the wavelength detection range of luminescent plate readers. Further, when developing NanoBRET assays, it is in fact recommended to use red and far red tracers, despite poor spectral overlap with Nluc, because red tracers result in lower background[9,11,43]. To overcome this issue, we have taken advantage of the distance between two different FFA1 binding sites, as well as the chemical properties of the fluorophores[27]. This has allowed us to develop DISCo-BRET where it is the distance between the Nluc energy donor and the final acceptor across the cell membrane that dictates the requirement for an intermediator ligand.

While we have employed this approach to study the FFA1 receptor, given that there are increasing numbers of structures showing GPCR ligands binding to extrahelical allosteric binding sites[8], we anticipate our approach can be adapted to study many other GPCRs. The allosteric site two in FFA1 is comprised of a pocket formed between TM3, TM4, and TM5, and importantly similar pockets have been identified for multiple other GPCRs, including dopamine, adrenergic and complement component C5a receptors[44]. Interestingly, there is also diverse pharmacology for ligands binding to this site, including agonists, as well as positive and negative modulators. An analysis of >500 GPCR structures found that all receptors had a pocket

in this location and that these pockets varied significantly in size and shape[45], suggesting this pocket is likely to be fruitful in future drug discovery. That said, we fully anticipate that DISCo-BRET can be further extended to study interactions between other GPCR binding site by identifying tracer ligands with appropriate chemical properties.

FFA1 is a clinically relevant GPCR with potential for the treatment of diabetes or other metabolic disorders[12,46]. There is also evidence that FFA1 ligands binding to these two sites have different biological effects[47]. By helping to understand how these sites interact with each other, we will open new opportunities for future FFA1 ligand development. Using DISCo-BRET we have uncovered FFA1 ligand pharmacology, specifically identifying site one antagonists with unique abilities to modulate function at site two. The fact that DISCo-BRET has allowed us to quickly identify these ligands in a single assay, provides a clear advantage over traditional approaches to characterise allosteric ligands[48]. Further, in examining the interactions between our site one and site two tracers, it was notable that while the binding kinetics was altered for the site one tracer, there was no effect on binding kinetics for the site two tracer. Given the structure of the FFA1 site one[14], and likely ligand entry through an opening accessed via the lipid, the altered kinetics we observe likely suggest that binding of our site two ligand closes off access through this opening, resulting in a reduction in both the on and off rate (Fig. 7).

DISCo-BRET provides a innovative technique to explore allosteric interactions between two ligands binding to the same GPCR at the same time. Allosteric GPCR ligands are an important area for future drug discovery, due to their ability to maintain spatial and temporal signalling, or to bias receptor signalling[49]. A central principle of ligands that allosterically modulate each other is that this interaction should be reciprocal[4]. It is therefore surprising that we observe a clear slowing of binding rates for our red site one tracer, but no clear effect on our green site two tracer. However, it is notable that while the on and off rates for the red tracer slow, the binding affinity does not actually change, suggesting perhaps that the green tracer is affecting access

into and out of the binding pocket, but not the binding site itself. Alternatively, the full agonist green tracer may be stabilising a G protein bound conformation of the receptor, which has been shown to slow orthosteric agonist binding kinetics for other GPCRs[50]. Future studies should aim to fully understand allosteric reciprocity between the two FFA1 binding sites.

In summary, we have developed an approach to measure simultaneous ligand engagement with multiple GPCR binding sites. We have used this to understand and develop ligands for FFA1, a clinically validated drug target. This approach will help provide key insights to the future of allosteric GPCR drug discovery.

# Methods

## Chemical synthesis
All compounds were synthesised as described in the Supplementary Methods. The identity of synthesised compounds was confirmed by NMR and HRMS and all compounds used were shown to have a minimum of 95% purity.

## Plasmids
A plasmid containing the FFA1 sequence fused in frame at its C terminal to Nluc was generated in the pcDNA5 FRT/TO (ThermoFisher Scientific, UK) plasmid backbone. The sequence for the human FFA1 receptor without its stop codon was inserted between HindIII and KpnI sites, while the Nluc sequence inserted between KpnI and XhoI sites in the parental plasmid. Plasmids encoding $\beta_2$-AR fused at its C terminal to Nluc and untagged FFA1 were generated through an enzyme free end homology cloning strategy[51].

The identity of the resulting plasmids were confirmed by sequencing (Eurofins).

## Cell culture and transfections
All Flp-In T-REx 293 cell lines (Thermo Fisher, R78007) were maintained in Dulbecco's Modified Eagles Medium (DMEM), supplemented with 10% foetal bovine serum, penicillin/streptomycin and normocin (InvivoGen, UK). Cells were cultured at 37 °C and 5% $CO_2$ in a humidified atmosphere and subcultured as required. The Nluc-FFA1 Flp-In T-REx 293 cell line, inducibly expressing the FFA1 receptor fused at its N terminus to Nluc, was as described previously[17]. The FFA1-Nluc Flp-In T-REx 293 cell line, inducibly expressing the FFA1 receptor fused at its C terminus to Nluc, was generated by first using polyethylenimine to co-transfect parental Flp-In T-REx 293 cells (ThermoFisher Scientific, UK), with the FFA1-Nluc plasmid and a POG44 plasmid (ThermoFisher Scientific, UK), expressing an Flp recombinase. Following transfection, cells were selected using blasticidin and hygromycin B to obtain an isogenic cell line with FFA1-Nluc incorporated into the Flp recombinase site. For all experiments with Flp-In T-REx 293 cell lines, cells were treated with 100 ng/mL of doxycycline for 24 h to induce expression of the transfected construct.

## Ca²⁺ mobilisation
Nluc-FFA1, FFA1-Nluc or untagged FFA1 Flp-In T-REx 293 cells were plated at 50,000 cells per well in poly-D-lysine coated black clear bottom 96-well plates. Cells were then cultured for 24 h until reaching confluency before culture medium was replaced with medium containing 100 ng/mL of doxycycline to induce receptor expression. Cells were cultured for a further 24 h before the assay. On the day of the assay culture medium was removed and replaced with medium containing 3 μM Fura2-AM Ca²⁺ dye. Cells were returned to the cell culture incubator for 45 min before medium and dye removed, and cells washed twice with Hanks Balanced Salt Solution supplemented with 20 mM HEPES (HBSS). After washing cells were incubated in HBSS for 15 min at 37 °C before being transferred to a Flexstation micro-plate reader (Molecular Devices). For experiments that involved antagonism, the antagonist ligand was added during this 15 min incubation.

Fura2 fluorescent emission at 520 nm following excitation with each of 340 and 380 nm was measured in regular intervals both before and after adding test compounds for a total of 90 s per well. Calcium responses were determined by taking the peak 340/380 ratio after first subtracting the ratios obtained prior to addition of test compound.

## NanoBRET
Nluc-FFA1 or FFA1-Nluc Flp-In T-REx 293 cells were plated at 50,000 cells per well in poly-D-lysine coated white 96-well plates. Cells were cultured for 24 h in the plate before medium was replaced with medium containing 100 ng/mL of doxycycline to induce receptor expression. After culturing for a further 24 h, medium was removed, cells were washed with HBSS and incubated in HBSS for 30 min. To start the assay, NanoGlo substrate (Promega, N1110) was added to the cells at a 1:800 final dilution and incubated for 10 min at 37 °C. For endpoint assays, including both saturation and competition binding, test ligands were then added, including both fluorescent and unlabelled competing ligands as indicated, and plates incubated for a further 5 min at 37 °C. NanoBRET was then quantified in a CLARIOstar microplate reader (BMG Labtech) by measuring luminescent emission with the monochrometer set for the Nluc donor at 450−60 nm, and acceptor luminescent emission 540−60 nm (NBD), 600−80 nm (Cy3.5) or 690−100 nm (SulfoCy5), depending on which fluorophore was being tested. The acceptor/donor BRET ratio was then taken as a measure of fluorophore binding. For kinetic on then off NanoBRET experiments plates were washed the same way and substrate was added to the same concentration. However, 10 min after substrate was added plates were inserted into the CLARIOstar microplate reader set to 25 °C to take repeated measurements of luminescent emission for both the donor and the acceptor fluorophore being used. After 5 min of background measurements, the indicated concentration of fluorescent ligand was added using the plate reader injector and luminescent emission was monitored for ~10 min before the reader was paused to manually add a high concentration of appropriate competing ligand to initiate dissociation. The plate was returned to the reader and luminescence measurements were continued for a further 30 min to monitor dissociation of the fluorescent ligand. For kinetic experiments using multiple concentrations of fluorescent ligand, cells were placed in the plate reader immediately after the addition of Nano-Glo substrate and luminescent measurements were taken. After 5 min, the reader was paused to add the indicated concentrations of fluorescent ligand, before measurements were continued. To set non-specific binding for kinetic experiments, the indicated competing ligand was added 5 min before the addition of the fluorescent ligand.

## DISCo-BRET
Nluc-FFA1 or FFA1-Nluc Flp-In T-REx 293 cells were plated at 50,000 cells per well in poly-D-lysine coated white 96-well plates. Cells were cultured for 24 h in the plate before medium was replaced with medium containing 100 ng/mL of doxycycline to induce receptor expression. After culturing for a further 24 h, culture medium was removed, cells washed with HBSS and incubated in HBSS for 30 min. For all DISCo-BRET formats Nano-Glo substrate was then added at a 1:800 final dilution and cells incubated for 10 min. Following substrate addition experimental set up different among the three formats: endpoint, spectra scan, and kinetics.

For endpoint experiments all fluorescent and unlabelled ligands were added to the cells and incubated for 5 min at 37 °C. Luminescent emission was then measured using a CLARIOstar microplate reader with monochromator set at 450−60 nm (Nluc), 540−60 nm (NBD) and 690−100 nm (SulfoCy5). BRET ratios were then calculated as either SulfoCy5 emission divided by NBD emission, SulfoCy5 emission divided by Nluc emission, or NBD emission divided by Nluc emission.

For spectra scan experiments all indicated fluorescent and unlabelled ligands were added to the cells and cells incubated for 5 min at

37 °C. Luminescence emissions were then measured from each well of the plate from 440 nm – 740 nm with a 5 nm resolution using a CLARIOstar microplate. Luminescent emission spectra were then normalised as a percentage of the maximal Nluc emission peak (~ 460 nm), then subtracted with the spectrum obtained in the presence of high concentrations of ligands competing for both binding sites to obtain specific BRET binding spectra.

In kinetic DISCo-BRET experiments plates were placed directly in a CLARIOstar plate reader set to 25 °C 10 min after Nano-Glo substrate addition. Luminescent measurements were then taken with three sequential measurements per well with the monochromator set to 450–60 nm (Nluc), 540 - 60 nm (NBD) and 690–100 nm (SulfoCy5) taken in 20 or 30 s intervals. After 5 min of background measurements, the plate reader was paused for the first indicated fluorescent ligand(s) addition, before immediately returning the plate to the reader to continue luminescence readings. After 5 min, the reader was again paused to add the second fluorescent ligand (association assays) or competing ligand(s) (dissociation assays), before immediately continuing the measurements for the indicated time. Non-specific control binding was obtained by adding the indicated competing ligand(s) 5 min prior to the addition of the first fluorescent ligand.

### Bystander BRET Internalisation
For internalisation assays, HEK293T cells (ATCC, CRL-3216) were co-transfected with either FFA1 or $\beta_2$-adrenoceptor constructs with Nluc fused to their C terminal and with an mNG fluorescent protein construct containing the CAAX domain from KRas. Cells were transfected using a reverse transfection protocol while plating into a poly-D-lysine coated white 96-well plates. Cells were plated at 60,000 cells with 100 ng of plasmid DNA and 0.6 μg of polyethyleneimine (PEI) per well. Cells were cultured for 24 h before washing with HBSS and incubating in HBSS at 37 °C for 30 min. NanoGlo substrate was added a final 1:800 dilution and plates incubated for 10 min at 37 °C. Luminescent emissions were then taken at 1 min intervals, simultaneously measuring 535 and 475 nm emission, using a PHERAstar FS microplate reader (BMG Labtech) set to incubate at 37 °C. After 5 min of baseline measurements, readings were paused and test compounds added to the plate. The plate was then returned to the reader to continue luminescent emission measurements for a further 55 min. BRET responses were quantified as the 535/475 nm emission ratio and were first corrected for the baseline ratio, before subtracting vehicle responses to obtain NetBRET.

### NanoBit arrestin-3 recruitment
For arrestin-3 recruitment assays, HEK293T cells were transfected with FFA1 receptor and a bystander NanoBiT split luciferase arrestin-3 recruitment biosensor. Cells were transfected and plated simultaneously in a reverse transfection protocol into a poly-D-lysine coated white 96-well plate. Per well, 60,000 cells were transfected with 5 ng plasmid expressing LgBiT with a N-terminal Lyn11 plasma membrane localisation tag, 5 ng plasmid expressing bovine arrestin-3 fused at its N-terminal to SmBiT tag and 30 ng plasmid expressing the human FFA1 receptor. Total DNA was made up to 100 ng total/well with empty pcDNA3 vector and transfected to cell suspension with 0.6 μg/well PEI. Following transfection cells were cultured for 24 h before use. On the day of assay, cells were washed twice with HBSS and incubated in HBSS for 30 min at 37 °C. NanoGlo substrate was added at a final 1:800 dilution and incubated in the dark for 10 min at 37 °C. Raw luminescence measurements were taken with 0.5 s read time and 1 min intervals for 25 min using a CLARIOstar plate reader (BMG Labtech), with serial dilutions of indicated test compounds added after 5 min. Raw luminescence values were divided by baseline signal (average of first 5 data points prior to compound addition) and vehicle measurements subtracted from agonist treatment. Area under curves were calculated and concentration response curves plotted.

### Data analysis
All data presented in graphs are mean ± SEM from the indicated number of experiments. For curve fit calculations of pEC$_{50}$, pIC$_{50}$, pK$_i$, E$_{Max}$ and percent competition, individual curve fits were carried out to derive these parameters from each independent experiments. The mean ± SEM of these parameters from each experiment was then calculated and reported in the data tables. Initial ratiometric data analyses were carried out for Ca$^{2+}$ assays using Softmax Pro v. 5.3 (Molecular Devices), or using MARS Data Analyses Software v. 3.20 (BMG Lab-Tech) for all other assays. Subsequent data analyses and curve fitting were carried out using Graphpad Prism v. 10.5 (GraphPad Software, USA). For all concentration response experiments, data were fit to a three-parameter log agonist concentration vs response model, where the vehicle treatment was plotted at one log unit lower than the lowest tested concentration. Saturation, displacement and kinetic binding data were fit to models as is indicated in the respective figure legends.

### Reporting summary
Further information on research design is available in the Nature Portfolio Reporting Summary linked to this article.

## Data availability
The authors declare that the data supporting the findings of this study are available within the paper and its supplementary information files. Source data are provided with this paper.

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

## Acknowledgements

This work was supported by the Lundbeck Foundation (ERU, grant no. R307-2018-2950), the Academy of Medical Sciences (BDH, grant no. SBF004\1033) and Tenovus Scotland (BDH, grant no S18-20). The work was also supported by the EPSRC and SFI Centre for Doctoral Training in Engineered Tissues for Discovery, Industry and Medicine, Grant Number EP/S02347X/1 (EMD); a BBSRC doctoral training partnership, Grant Number BB/X010902/1 (OL); and a Medical Research Scotland Studentship (LV and BDH, PHD-50838-2024). We would also like to recognise Matjaz Brvar and Christoffer V. Heidtmann for help with synthesis of

fluorescent building blocks, and Junsheng Chen and Lasse Brokmose Poulsen for help with recording fluorescence spectre.

## Author contributions

The project was conceived by ERU, and funding secured by ERU and BDH. AV and AM supervised by ERU, and MC, supervised by ERU and TU, carried out all chemical synthesis and characterisations. All biological cell-based assays and their analyses were designed and supervised by BDH. These biological assays include: $Ca^{2+}$ assays carried out by AV, BD, BS, BDH; NanoBRET binding assays carried out by AV, BD, OL, CM, BDH; arrestin-3 recruitment assays carried out by EMD; internalisation assays carried out by LV, SC; and DISCo-BRET assays carried out by BDH. The DISCo-BRET name was conceived by EMD and BDH. The original draft of the manuscript was written by BDH and all authors contributed to editing of the manuscript.

## Competing interests

The authors declare no competing interests.
