## [Peer Review file · Nature Communications]

Multicolored sequential resonance energy transfer for detection of simultaneous ligand binding at G protein-coupled receptors

Corresponding Author: Dr Brian Hudson

Version 1:

Reviewer comments:

Reviewer #1

(Remarks to the Author)

The manuscript by Valentini et al. describes the development of a novel approach termed multi-coloured Dual Interaction Sequential Compounds (DISCo)-BRET enabling to measure simultaneous ligand engagement to multiple binding sites at the G protein-coupled receptor (GPCR) FFA1. For their approach, the authors developed two fluorescent ligands binding to two distinct binding sites at FFA1 and used these fluorescent ligands in combination with the NanoBRET technology. The fluorescent ligand targeting FFA1 site II was developed starting from a previously identified chemotype (i.e., Cpd 6h). The development for the fluorescent ligand for FFA1 site I was based on a previously reported fluorescent ligand targeting this binding site (i.e., TUG-1460). By a careful selection of the used fluorophores in combination with an attachment of the Nluc to the intracellular C-terminus of the receptor, the authors were able to achieve sequential resonance energy transfer from the intracellular Nluc to the site II fluorescent ligand (i.e., TUG-2597) and finally to the site I fluorescent ligand (TUG-2591). In competition binding studies this approach was applied to map known FFA1 ligands to the two different binding sites of the receptor. Moreover, the established assay setup enabled to detection of cooperativity between ligands binding to site I and II and was used for kinetic binding studies.

In my opinion, the described DISCo-BRET technology will be highly relevant for future drug discovery studies targeting FFA1. Moreover, the developed DISCo-BRET technology can be transferred to other GPCRs with more than one known ligand binding site. Moreover, the manuscript is well-written and well-structured. Therefore, I suggest this manuscript to be published in Nature Communications. However, following (major and minor) points should be addressed prior to publication.

Major points:

- The authors are highly encouraged to apply their DISCo-BRET technology beyond the presented proof-of-principle study. An application of this technology for the identification and characterization of novel FFA1 ligands with improved affinity and/or activity would clearly show how this new concept can leverage drug discovery in the field of GPCRs, thus clearly rationalizing a publication of this study in a high ranked journal such as Nature Communications.

Minor points:

- Abstract: The target receptor (i.e., FFA1) should be mentioned in the abstract.
- Figure 1A: If the racemate or one of the two enantiomers of TUG-2489, TUG-2490, TUG-2597, TUG-2598 was used, this should be clarified.
- Figure 1C: Which radii were used for the illustration of the green and blue spheres? Site one and site two might be labelled at the receptor.
- It would be nice, if the structure of TUG-1460, which was used as a starting point for the development of TUG-2591, could be shown in the manuscript.
- Discussion: The statement "..., to date it has not been possible to use NanoBRET to study binding to multiple sites on the same receptor, at the same time." is not fully correct. In recently reported work (Huber et al. ChemMedChem, 2024, e202400284; Vogt et al. ACS Pharmacol Transl. Sci., 2024, 7, 1533-1545) NanoBRET based assays were used to detect binding to an intracellular allosteric binding site (by direct competition with an intracellular fluorescent ligand targeting this site) but also allowed to study binding to distinct extracellular binding sites as a consequence of cooperative effects. Therefore, the statement mentioned above should be modified.
- Reference list: For some references the page numbers (i.e., 16, 18, 30) are missing.
- Suppl Figure 1. What is the difference between the conditions TUG-2597 + T360 (orange triangle) and TUG-2597 + T360

(red rectangle)?

- Suppl Figure 1 Aiii: The “.” in “TUG-2597 + T360. + TUG-905” should be removed.
- Some synthesis schemes in the Supporting information (e.g., pages 15, 17, 41) are still shown in a “track changes mode” with lines at the left side and the bottom of the scheme. The same is true for the ¹H NMR of TUG-2598 (Figure S14)
- Supplementary References: For some journal names the abbreviation is used (e.g., Angew Chem Int Ed Engl) for other not (e.g., Journal of Medicinal Chemistry).

Reviewer #2

(Remarks to the Author)

Valentini et al. described three-color FRET, called “Dual Interaction Sequential Compound (DISCo-BRET).” A technique based on Bioluminescence Resonance Energy Transfer (BRET) was used to unravel the binding kinetics of two allosteric ligands to the Free Fatty Acid Receptor, FFA1. The authors spent time and effort synthesizing fluorescent compounds used in the assay. In my opinion, this is a method paper developed to study ligand binding to previously characterized binding sites to FFA1. Some effects are too small to convince and backed by enough evidence. Also, the manuscript lacks proofreading because there are too many errors, mainly the ligand names. Below are some major and minor comments on the manuscript.

Major comments:

1. There is no supporting data provided in the manuscript that validates the luciferase (Nluc)-FFA1 fusions at both the N- and C- terminus are active (Ca²⁺ mobilization assay). While the generation of N-terminus fusion is referenced (Reference 16), the data demonstrating the functionality of the Nluc fusions of FFA1 compared to the unmodified (wild-type) FFA1 is absent in both the referenced article and the current manuscript. I highly recommend that the authors include this validation data in the current manuscript and direct readers to the relevant references. Without this, the claims regarding the functionality of the Nluc-FFA1 fusions remain unsubstantiated, and their similarity to the WT FFA1 is not confirmed.
2. In Figures 2 to 4, the competitive ligand TUG-2193 is mentioned. Similarly, another ligand named TUG-2456 is noted in Figure 4. However, these ligands were not explained in the text and the supplementary information, and no proper referencing or structures were also provided. Also, in the results section titled, ‘Allosteric Regulation of FFA1 Ligand Dissociation’ and the corresponding supplemental information, the TUG-2193 and T360 were interchangeably used in the figures, text, and supplemental information. It is unclear which of these compounds were used for the competition assay.
3. The choice of concentrations of compounds for the competition assays is not justified and well explained.
4. The Nluc in the C or N-terminus is shown to excite the terminal acceptor Sulfo-Cy5 along with the NBD dye, and the authors claim this energy transfer is not as effective as that of the NBD, as shown in Figure 3. However, in Figures 5F and 5G, when the authors use the single ligand NanoBRET format, the ‘Specific BRET’ between Sulfo Cy5/Nluc is much higher (~1 to 4) than that of NBD/Nluc (~0.01 to 0.04). The difference here is not explained well in the text, and the references to these figures in the Result section are wrong. Are both the green and red tracers still present in these experiments? If so, can the authors try an unlabeled site-one ligand as a control to verify this observation? For example, can the authors premix the green tracer (TUG-2597) with compound 26 in the synthesis steps of the red tracer (TUG-2591) before the competition assay with T360? In this way, one can directly observe the BRET values decreasing from the dissociation of the green tracer without the energy transfer to the red tracer and verify if the dissociation rate is similar to Figure 5F.
5. In the Discussion section, lines 384 to 387, the authors mention that the ligand entry to site-one via the lipids impacts the binding of the site-two ligand. Can the authors explain the effect in more detail? Does the binding of the site-two ligand open or close the site-one? Overall, what is their hypothesis of the ligand binding at the site two effects on the site one based on their observations? Can the authors add a conclusion figure as a cartoon to summarize based on their binding kinetics?
6. Labeling infra-red dye like Alexa750 or Cy7 would not overlap with Nluc emission, but overlapping only with NBD emission would significantly strengthen the conclusions made by the manuscript.
7. Data analysis was not explained adequately; it only mentioned that all data presented were carried out using Graphpad Prism 10 (GraphPad Software, USA).

Minor comments:

1. In the Introduction, lines 65 to 66, the sentence is unclear, hard to understand, and must be rephrased. It also applies in the results section, lines 111-113.
2. Does ‘Cpd 6h’ bind to sites one and two? This compound is listed in both the Supplementary Tables 1 and 2. Can this be explained?
3. Supplementary Table 2 is not referred to in the Results section 2, and Supplementary Table 1 is referred to instead.
4. In Figure 2, which details the development of the red tracer to orthosteric site one, the Ca²⁺ mobilization data were normalized with the binding of T360. However, it is mentioned as a site two ligand. Can the authors clarify in the text why they chose T360?
5. Line 240-241; “In contrast, the site two compounds, T360 and Cpd 6h, display clear competition in both DISCo-BRET and green tracer binding (Fig. 4Aii and 4Bii. I didn’t see any of these compounds were tested here. Also, there is no effect of aLA, but the authors mentioned it as a modest effect.
6. In Figures 3C & 3D, the same symbol has been used to represent the Specific BRET values for the conditions 100 nM TUG-2591 and control (TUG-905/TUG-2193). It needs to be fixed.
7. In Figure 5, it is hard to distinguish the blue and green curves. I recommend that the authors try different symbols for them.
8. The caption for Figure 2B and the corresponding Results sections detail using the ligand Cpd 6h for Ca²⁺ mobilization assay. However, the Cpd 6h data is not shown or mislabeled in Figure 2B.

Reviewer #3

(Remarks to the Author)

In the manuscript entitled "Multi-coloured sequential Resonance Energy Transfer for simultaneous ligand binding at G protein-coupled receptor" written by Valentini and collaborators, the authors aimed at investigating the interaction between the binding site of various ligands and more specifically the interactions between different allosteric binding sites.

The authors developed an original technical approach based on sequential RET between a receptor (Free Fatty acid Receptor (FFA1)) and fluorescent ligands labeled with NBD and SulfoCy5 and specific for the allosteric binding site 1 or 2.

General consideration

As mentioned above, the approach is original and, whereas the various attempts at sequential RET strategies developed to study receptor oligomerization, for example, are not convincing, the demonstration of sequential RET is here rather well done.

But the overall impression is that the manuscript suffers from numerous inconsistencies and approximations: the panels in the various figures do not correspond to the description given in the manuscript, some ligands do not have the same name in the text and in the figures, references to the figures in the text are not well done and the reader has to find the right panel corresponding to the text. Values for ligand affinity, K_{off} and K_{on} are given without error (SD or sem) except in Suppl. Table 3. Finally, the conclusions drawn from the results are not discussed.

From a scientific point of view, as mentioned above, the strategy developed may be interesting, but it doesn't allow to consider interactions between sites. It is simpler to perform two BRET measurements with each of the tracers. The authors mention that the identification of allosteric modulators is a major focus of GPCR research, but Disco-BRET is too complicated to finely characterize new molecules. Finally, the benefits of DISCo-BRET compared with Nano-BRET are unclear. It seems much easier to perform and interpret Nano-BRET experiments with the Nluc-FFA1 receptor and each of the two tracers separately than to perform Disco-BRET.

Indeed major questions have to be answered before using such a technique, especially to investigate the interactions between binding sites:

- The proportion of receptors labeled with each of the two tracers and with the two tracers simultaneously: in the present study the concentrations of the tracers are not compatible with a saturation of the two binding sites. Therefore, the authors have to deal with a mixture of the different complexes and the values for the affinity, K_{off} and K_{on} correspond to the mean of the values of the different species.
- In the disco-BRET configuration, the red signal and the green signal are both dependent on the binding of both tracers. A decrease of the occupancy of site 2 can induce a decrease of the red signal but it is not necessarily due to a decrease of the occupancy of site 1. Conversely, a decrease of the occupancy of site 1 can induce an increase of the green signal without any increase of the occupancy of site 2. It means that it is almost impossible to correlate the variation of the signal with the occupancy of the binding sites.
- The addition of a third ligand as a potential competitor leads to an even more complicated situation. Since positive or negative cooperativity are dependent on the ligands themselves, the final result of one experiment is dependent on the cooperativity between the tracers and between the unlabeled ligand with the tracer. For example, the addition of TUG-905 to displace TUG-2591 modifies the cooperativity between TUG-2591 and TUG-2597, and that was what the authors were looking for, but the binding of TUG-905 introduces a new type of cooperativity between TUG-905 and TUG-2597. Therefore, the variations of the green signal are dependent on both cooperativities which are not necessarily the same.
- BRET signal variations in the manuscript have always been associated to a variation of the occupancy of the binding sites but the binding of a ligand, especially agonists as done in the present study, can have a deep impact on the structure of the receptor. This leads to variations in the distance between the Nluc and the tracers and therefore to variations of the BRET signals. This point has never been discussed or even considered in the manuscript.
- The authors have used fluorescent agonists. Did they evaluate the potential consequence of the binding of the tracer on receptor internalization?
- The discussion on the reciprocity between the binding sites is not accurate.
- Finally, the authors should have discussed the interest of the strategy for other GPCRs. It is most probable that the DISCo-BRET can be relevant only for receptors for which a binding site equivalent to the binding site 2 (i.e. located deep inside the receptor) has been described.

The authors will find below different points which can refer to the points mentioned above.

Specific points

- 1- The manuscript is not easy to read and suffers from inconsistency. Throughout the manuscript, the authors mentioned the ligands T360 and Cpd 6h and referred to different figures and tables. But none of these two ligands were mentioned in the figures. For example, lines 240-243: "In contrast, the site two compounds, T360 and Cpd 6h, display clear competition in both DISCo-BRET and green tracer binding (Fig. 4Aii and 4Bii). It is notable that these compounds did not result in complete competition of the DISCo-BRET (Fig 4. Aii; Sup. Table 2)". We ended up assuming that T360 and Cpd 6h correspond to TUG 2193 and TUG 2456 but without being completely sure.
- 2- In the introduction, the authors wrote "... with studies demonstrating at least two distinct allosteric sites on the receptor" (line 87). It is not clear why these binding sites have been called allosteric and not orthosteric? Moreover, usually allosteric ligands do not promote receptor activation (except the ago allosteric ligand) but, in the present study, the authors have selected the green and red tracers for their capabilities to activate fully the receptor.
- 3- Most of the values of the potency or affinity are indicated without errors (example line 130, Figure 1E or F). Please provide SD for all the values. Also indicate for all the graphs if the plots correspond to pooled data from independent experiments or to a "representative" experiment.
- 4- line 184 : "... that incorporating physical separation ..." can give the idea of a physical barrier. "increasing the distance between Nluc and the red tracer, TUG2591, ..." is maybe better.
- 5- Figure 3: it would have been better to specify on all the panels what "specific BRET" means, as done in Figure 4.
- 6- Why are the legends of Y-axis for the Disco-BRET in Figure 3 E and F (Disco-BRET (700nm/525nm)) and in Suppl Figure 1 (Specific BRET 690/540) not the same? Is there a difference in the ratio? If so, why?
- 7- Are the legends of suppl Figure 1 Aiii and Biii correct? It seems that TUG-2597 + T360 corresponds to two symbols but

TUG-2597 + TUG-905 is not present.

-8- There is no mention of T360 and Cpd 6h in Figure 4 and no mention of TUG 2193 and TUG 2456 in the text of the manuscript. Therefore, it is difficult to follow the argumentation.

-9- In the description of the results illustrated in Figure 4 Ai and Bi, there is no comment on the variation of the slope of the curves. In panel Bi, the slope of the curve can be greater than 1 (TUG-905) and lesser than 1 (TUG-770 and Cpd-B) suggesting potential positive or negative cooperative interactions between the binding sites 1 and 2, depending on the competitor.

Because in Panel Ai, the BRET signal corresponds at the same time to the presence of the red tracer but also the green tracer, one would expect to see variation in the slopes in the competition curves, as in Panel Bi. It is not the case. The authors have to comment such discrepancies. Moreover, it is mentioned in the "Data Analysis" section that data were fitted to a three-parameter log agonist concentration vs response model. It is obviously not the case in Figure 4 Bi.

-10- lines 250-251: "that both T360 and 251 Cpd 6h actually have positive cooperativity with the red tracer (Fig. 4Ci; Sup. Table 3)". It could be the case but one can also imagine that the increase in the BRET signal induced by two compounds in Panel Ci is due to a structural modification of the receptor leading to shorter distance between Nluc and the red tracer. The authors have to discuss that point, even if it is less probable.

-11- The authors developed a sequential RET system to study the interactions between the binding sites 1 and 2. But one of the stronger evidence of interactions between the binding sites is provided by Figure 4 Ci and Cii when using the Nluc-FFA1 receptor and measuring the BRET with the two tracers independently. Of course, it needs two experiments but the interpretation is much more reliable. Therefore, is it necessary to develop a sequential RET assay to study binding site interactions or are the investigations on the interactions between the binding sites the best model to prove the interest of Disco-BRET?

-12- lines 264-265: "Tracer binding was monitored as either DISCo-BRET to assess binding of the red tracer (Fig. 5B)". The formulation of the text is not correct since Disco-BRET is dependent on the binding of both tracers. "Tracer binding was monitored as either DISCo-BRET to assess simultaneous binding of the green and red tracers (Fig. 5B)" would be more accurate.

Moreover, in Figure 5, TUG 2591 and TUG 2597 were used at 100 nM and 316 nM, respectively. At these concentrations, when considering the affinity reported in Table 4, respectively 250 and 110 nM, we can estimate that about 30% of site 1 and 75% of site 2 are labeled. It means that some of the receptors are empty, others bound to one of the two ligands and a small proportion to the two ligands (~23%, if we do not consider any cooperativity). Therefore, it means that when using the Disco-Bret (BRET 690/540) and the nanoBRET (540/450), the authors dealt with different types of receptors depending on whether they are linked to one (TUG 2591 or TUG 2597) or two ligands (TUG 2591 and TUG 2597). In these conditions, it is difficult to compare the signals.

-13- lines 268-270: "It was notable that the off-rate for the red TUG-2591 from site one was substantially slower than that of the green TUG-2597 from site two." Please indicate the figure to be referred to. Moreover, if we consider values reported in Suppl. Table 4, it is not obvious, the K_{off} are respectively 0.41 and 0.38 for TUG-2591 and TUG-2597, respectively. Finally, in Figure D, the decrease of the signal can either be due to the dissociation of the red or the green tracer.

-14- Figures 5 F and G do not correspond to the legend beneath. It seems that panels have been inverted. And the different panels referred to TUG 2193 while the legend and the manuscript referred to T360. All these approximations make the reading of the manuscript very difficult! Moreover, it would have been very interesting to use the same protocol used in Panel F and G and to follow the dissociation of TUG -2591 and TUG 2597 in the presence of an excess of T360 and TUG-905, respectively.

-15- Lines 293-296: "When assessing site two ligand binding, association was observed after addition of green TUG-2597 alone, which was enhanced by the presence of TUG-905, consistent with positive cooperativity between these ligands." It is not clear to which panel the text referred, please indicate the panel to be considered. If it is Panel C, the conclusion is hasty. Various hypotheses can be formulated. First, nothing indicates an increase of the association but simply an increase of the signal. Because the red tracer is displaced by TUG-905, there is no more FRET from the green tracer to the red tracer and therefore the fluorescence of NBD increased. Second, there is no proof of cooperativity.

Moreover, and by contrast to what panel A and D can suggest, the proportion of complexes receptor/green tracer/red tracer is not known since the concentration of the tracers used do not allow a saturation of the binding sites. This is a major default of the study.

-16- lines 308-310: "As before, binding of green tracer was measured using the NBD/Nluc BRET ratio (Fig. 6F), while binding of the red tracer was assessed through DISCo-BRET and SulfoCy5/NBD ratio (Fig. 6E)." Again, panels 6E and 6F as described in the text do not correspond to the panels in Figure 6 ... probably another inversion!

-17- lines 313-314: "These results demonstrated a rapid binding of green tracer, which was not impacted by the addition of red tracer, although green tracer binding was enhanced by pre-treatment with the site one competing compound, TUG-905 (Fig. 6E). Addition of red tracer did however lead to a clear increase in DISCo-BRET, but notably this occurred with a relatively slow association rate (Fig. 6F; Supp. Table 5)."

If the complex receptor/green tracer/red tracer is one of the major species in the preparation, the addition of red tracer should necessarily impact the NBD/Nluc signal. The absence of variation could reflect a low proportion of receptor/green tracer/ red tracer complex.

Requested experiment:

1) saturation binding curves (equivalent to Suppl Fig. 1) and kinetics curves (equivalent to Fig. 5D/E) for TUG-2591 (red) in DISCo-BRET mode, with for each a plateau at high concentration

2) The effects of the competitors have to be investigated on both binding sites in Naono-BRET when using Nluc-FFA1 receptor. The authors have to be sure that receptor occupancy is high enough to favour one complex species (at least occupancy of more than 75% of the binding site occupancy, i.e. at least $3x K_d$) : i) TUG-2597 (green) 316 nM alone, followed by TUG-905 high concentration to clearly establish TUG-905 effect on TUG-2597 ; ii) TUG-2591 (red) 750 nM alone,

followed by T360 high concentration to clearly establish T360 effect on TUG-2591

3) Do Fig. 5 B/C/E/F experiments again with similar receptor occupancies for each tracer, preferably with ~90% occupancy (concentration = 9 x Kd).

Conclusion : we consider that the manuscript can be published in its current state.

Reviewer #4

(Remarks to the Author)

The manuscript by Valentini et al., "Multi-Coloured Sequential Resonance Energy Transfer for Simultaneous Ligand Binding at G Protein-Coupled Receptors," is a very interesting study where an attempt is made to observe the simultaneous binding of several ligands to the receptor. The topic addressed in the publication is very important to the field – novel ligand binding assays to G protein-coupled receptors are needed to gain further insight into the pharmacology of these proteins. The focus of this publication is the NanoBRET assay, which has gained momentum in recent years and has become increasingly popular among the GPCR community. In that regard, the reference list should reflect the publications of different workgroups working on the subject. The developed DISCo-BRET assay is a promising approach for performing multiligand binding study. However, the manuscript in its present form raises several considerations which need to be addressed before it is ready for publication:

Fundamental:

1. The proposed method and approach are very elegant and interesting. However, avoiding bleed-through to red fluorophores from the Nluc energy donor is very challenging, especially when multiple fluorophores are involved. Although "physical separation" has been mentioned, no data are available to rule out potential optical interference. The description of physical separation described (lines 187-194) has not been confirmed by experimental data.
2. If FRET between ligands is indeed occurring, it should also be observed in wild-type receptors when the corresponding light or laser activates the green fluorophore. This would also exclude the possible effect of Nluc on the properties of the receptor. This kind of control experiment would give additional proof of multiple ligand binding.
3. It is unlikely that the classification of FFP1 allosteric ligands by efficacy is as straightforward as described in the publication ("Since site two FFA1 ligands behave as full agonists in G α q signaling, while site one ligands are partial agonists..."). It is widely known that the classification of compounds as full or partial agonists is highly dependent on the assay, expression levels and other used parameters making the results inconsistent. The publication gives the impression that site two targeting ligands cannot be partial agonists.

Major:

4. The abstract contains minimal information. There is no indication of the receptor that has been studied nor is the essence of the method described.
5. The binding and potency parameters in the text and kinetic parameters in the Supplementary Tables are given without confidence intervals. It makes it difficult to evaluate the significance of the obtained results and the correctness of the conclusions. Also, it is not written what the error bars on the figures correspond to (SD or SEM?) and whether the data are pooled or from single experiments.
6. T360 and Cpd 6h have been used as site two ligands (lines 240 -), but no data about these ligands have been presented in Fig. 4. Instead, the figure has data of ligands not described in the text. Fig. 2B text also indicates that the graph should have data of Cpd 6h, but there is no corresponding curve.
7. The dissociation rate constant depends on the extent of the dissociation (span). As it is not the same for all processes (Fig. 5), it is important to present these data and consider them while interpreting results.
8. The developed assay system could not detect the effect of the antagonist GW1100 on the fluorescent ligand binding. To confirm that GW1100 interacts with this receptor and behaves as an antagonist, experiments with Ca²⁺ mobilization would have to be performed. Only then can we make some conclusions about the localization of the antagonist binding site.
9. Fig. 2 F and G indicate substantial BRET connected with the binding of purple ligands to C-terminal Nluc. The presentation of these binding parameters would be important for the interpretation of the obtained results.
10. The modulation of fluorescent ligands' binding was also studied in the presence of the site one ligand TUG-905 (Fig. 6), but there is no information on how the pretreatment with site two ligands would alter the binding.
11. There are indications that some data is excluded from the graphs. For example, Fig. 2D is missing value of -10, Fig. 2F has eight datapoints, while Fig 2G-H have seven, Fig 3F has no 17.8 nM datapoint. The readers would appreciate all the measured data to be displayed on figures.
12. Were the association curves on Fig. 6B-C, E-F measured longer? The curves look like the association is not complete.
13. Is 10 μ M concentration of unlabeled ligand T360 (pIC₅₀ = 7.1-7.4) enough to saturate nonspecific binding in the presence of 3 μ M TUG-2597? Usually, great excess of an unlabeled ligand is used to determine non-specific binding.

Minor:

14. Data in Fig. 1E and F repeat (in normalized form) data presented in Fig. 1G and is an unnecessary duplication.
15. Fig. 4 would be easier to follow if the ligands are depicted in unique colours and the TUG-905 reference curve would be included with the datapoints.
16. Some conclusions about the efficacy of compounds are slightly misleading. For example, Fig. 1B compound TUG-2490 does not reach the same level of activation as other full agonists.
17. There is no Fig. 5H (line 280).

Reviewer #5

(Remarks to the Author)

Reviewer #6

(Remarks to the Author)

Version 2:

Reviewer comments:

Reviewer #1

(Remarks to the Author)

All my concerns have been addressed to full satisfaction. The manuscript can now be published in Nature Communications

Reviewer #2

(Remarks to the Author)

The authors have addressed most of the comments in the revised manuscript. I have few issues to be addressed.

In response to comment #6, the authors discuss potential improvements in Nano-BRET assays and plate readers to accommodate far-red dyes. I recommend incorporating this information into the third or final paragraph of the discussion section.

Minor comment:

In reference ID 25, there is an error displaying "INVALID CITATION."

Reviewer #3

(Remarks to the Author)

Thanks to the authors for addressing our comments. Much has been done and the manuscript has been much improved. We have only few requests concerning the second version of the manuscript.

Major points

-1- The absence of reciprocity: the green tracer has an impact on the red tracer but, conversely, the red tracer has no impact on the green tracer link. This phenomenon of non-reciprocity was addressed in the first version. Our comment in the previous review may have suggested that this aspect was not important. Perhaps we worded it wrongly, but it is an important aspect, but the explanations provided in the first version did not seem convincing to us.

How the authors explained this absence of reciprocity? They have to discuss this point in depth.

-2- The authors never considered the fact that the emission of TUG-2597 should change depending on whether or not there is an acceptor (TUG-2591) nearby. Indeed, it should decrease when TUG-2591 was added (Figure 6E) and the competition with unlabelled competitors for TUG-2591 (TUG-905 for example) should induce an increase of the emission (Figure 4Bi). We asked the authors to discuss that point and not to consider only "either cooperativity or a conformational change resulting in enhanced BRET between the C terminal Nluc and the green tracer". (line 242-244). The authors should discuss why displacement by unlabelled competitors of TUG-2591 bound to the receptor and close to the donor (TUG-2597) is not a plausible hypothesis to explain the increase in the specific BRET ratio. They also should explain why the addition of TUG-2591 in Figure 6E did not induced a specific BRET ratio decrease.

To our opinion, taking into account the % of receptors occupied concomitantly by TUG-2591 and TUG-2597 should provide part of the answer

-3- Validation of the model of ligand binding: to validate the model, we ask the authors additional experiments in BRET and compare the results to those already obtained in DISCO-BRET:

- 1) BRET Kinetics of saturation on Nluc-FFA1 with the same format than the one of Suppl 9A
 - o Labelling with TUG-2591 then addition of T360 and later addition of TUG-905(dissociation)
 - o Labelling with TUG-2597 then addition of TUG-905 and later addition of T360 (dissociation)

2) Kinetics of saturation on FFA1-Nluc: labelling with TUG-2597 and addition of TUG-905 and later addition of T360. This experiment is going to be useful to determine the effect of non-fluorescent site 1 agonists on NBD/Nluc ratio using the construct used for Disco-BRET and to discuss the possible conformational change, and its effect on non-visible allosteric reciprocity.

-4- The authors indicated that they had mistakenly used a 'two binding sites' equation to fit the curves in Figure 4Bi in the

first version and had replaced it with a 'one binding site' equation. Nevertheless, two binding sites must be considered, one (site 2) being an allosteric binding site for site 1. No adjustment procedure is available in Prism to fit their data, but perhaps the authors can clarify that the 'one binding site' fitting does not fully describe reality.

Minor points

- Suppl. Fig. 2: Y axis titles seem to be wrong: "% of untagged FFA1R maximum response" seem to be more accurate.
- Line 407-412: "physical separation » again => « substantial distance" could be better.

Reviewer #4

(Remarks to the Author)

The authors have addressed most of the issues reported by the reviewers and performed additional experiments, making the publication clearer and more interesting. The confusion with the ligand names has been resolved with correct labeling. However, there are still a few points that need to be considered before the manuscript can be published:

1. In Supplementary Figure 2A, TUG-905 is shown as a full agonist in the Ca²⁺ response (100% of T 360 response) with a pEC₅₀ close to 7.5 for wild-type receptors. This is inconsistent with other statements and data (Supplementary Table 1) in the manuscript regarding this ligand. The partial agonism appears only when the NLuc motif is coupled with the receptor (Supplementary Figure 2A). The impact of this behavior within the model needs to be analyzed.
2. It is good that the manuscript mentions that the efficacies of TUG-2489 and TUG-2490 cannot be fully determined (line 126, Figure 1B); however, this should also be noted in Supplementary Table 1.
3. The statement "mean ± SEM of pooled data from the indicated number of experiments" is still confusing. For pooled data, the value is not the mean but the best-fit value, and the standard error indicates the goodness of fit. This does not indicate the reproducibility of the experiment, which is usually required in this type of study.
4. The calculation of kinetic parameters in Figures 5D and 5E is still confusing. The "one site exponential decay model" used here contains three parameters: starting point, ending point, and rate constant. All these parameters change and influence each other. Therefore, presenting only the rate constant does not fully describe the processes. Additionally, alternative multisite models may be more feasible in some of the cases studied and should be considered.
5. Figures 2F-H describe the BRET signal of TUG-2591, TUG-2355, and TUG-2287 binding to C and N terminal NLuc constructs. Analysis data have been provided only for the N terminal construct. Since the BRET signal of TUG-2591 for the C-terminal construct is almost at the same level as the BRET signal of TUG-2355 for the N-terminal construct, it should also be evaluated. The statement "very little specific BRET" on line 176 is not relevant here.

Reviewer #5

(Remarks to the Author)

Reviewer #6

(Remarks to the Author)

Version 3:

Reviewer comments:

Reviewer #3

(Remarks to the Author)

The authors have responded to our various criticisms, and we consider that it can be published in Nature Communications

Reviewer #4

(Remarks to the Author)

Thank you for the additional data, clarifications and comments.
At present stage I think that the manuscript is already ready for the publication in the Nature Communications.

Reviewer #5

(Remarks to the Author)

Reviewer #6

(Remarks to the Author)

Reviewer #1 (Remarks to the Author):

The manuscript by Valentini et al. describes the development of a novel approach termed multi-coloured Dual Interaction Sequential Compounds (DISCo)-BRET enabling to measure simultaneous ligand engagement to multiple binding sites at the G protein-coupled receptor (GPCR) FFA1. For their approach, the authors developed two fluorescent ligands binding to two distinct binding sites at FFA1 and used these fluorescent ligands in combination with the NanoBRET technology. The fluorescent ligand targeting FFA1 site II was developed starting from a previously identified chemotype (i.e., Cpd 6h). The development for the fluorescent ligand for FFA1 site I was based on a previously reported fluorescent ligand targeting this binding site (i.e., TUG-1460). By a careful selection of the used fluorophores in combination with an attachment of the Nluc to the intracellular C-terminus of the receptor, the authors were able to achieve sequential resonance energy transfer from the intracellular Nluc to the site II fluorescent ligand (i.e., TUG-2597) and finally to the site I fluorescent ligand (TUG-2591). In competition binding studies this approach was applied to map known FFA1 ligands to the two different binding sites of the receptor. Moreover, the established assay setup enabled to detection of cooperativity between ligands binding to site I and II and was used for kinetic binding studies.

In my opinion, the described DISCo-BRET technology will be highly relevant for future drug discovery studies targeting FFA1. Moreover, the developed DISCo-BRET technology can be transferred to other GPCRs with more than one known ligand binding site. Moreover, the manuscript is well-written and well-structured. Therefore, I suggest this manuscript to be published in Nature Communications. However, following (major and minor) points should be addressed prior to publication.

Major points:

- The authors are highly encouraged to apply their DISCo-BRET technology beyond the presented proof-of-principle study. An application of this technology for the identification and characterization of novel FFA1 ligands with improved affinity and/or activity would clearly show how this new concept can leverage drug discovery in the field of GPCRs, thus clearly rationalizing a publication of this study in a high ranked journal such as Nature Communications.

-We thank the reviewer for this insightful comment. To demonstrate the utility of DISCo-BRET we have now employed the approach to define complex pharmacology and SAR of novel FFA1 antagonist ligands. Specifically, we screened a previously described FFA1 antagonist, PPTQ and a series of novel derivatives (TUG-2743-2745) in a DISCo-BRET competition assay (New Figure 4Aiii and Biii). These experiments showed that while PPTQ showed competition for both site 1 and site 2 ligands in the DISCo-BRET assay, some of its close chemical analogues display competition only with the site 1 ligand. We follow up these studies with functional antagonism studies (New Sup Figure 6) to confirm that while PPTQ can inhibit both site 1 (competitively) and site 2 (non-competitively) agonists,

the chemical analogues shown in DISCo-BRET to only compete with site 1 inhibit only site one agonists. To further confirm these observations we generate a novel green tracer antagonist (TUG-2884) based on these site one competing analogues, showing that this tracer binds to FFA1 (New Sup Figure 6), and that it shows competition only to other site one ligands (New Figure 4D). Together these data demonstrate the power of using DISCo-BRET to use a single assay to assess how novel ligands interact with ligands binding at different sites on the receptor.

Minor

points:

- Abstract: The target receptor (i.e., FFA1) should be mentioned in the abstract.

-Reference to the FFA1 receptor has been added to the abstract (Line 28).

- Figure 1A: If the racemate or one of the two enantiomers of TUG-2489, TUG-2490, TUG-2597, TUG2598 was used, this should be clarified.

-The compounds were synthesized as racemates and used without enantiomeric purification, in medicinal chemistry this is the expectation unless the structure or text indicates otherwise.

- Figure 1C: Which radii were used for the illustration of the green and blue spheres? Site one and site two might be labelled at the receptor.

-The illustration was created based on pictures exported from pymol and the spheres with soft edges are made to highlight the possible overlap between Nluc and NBD - so it cannot be specified exactly. However, based on the length of the receptor the green sphere has a diameter of 65 Å according to the measuring tool in pymol and the blue is app. 10% larger. Both spheres are slightly oval shaped. Since the figure is only made as an illustration we do not believe it important to include these details in the manuscript. We have added labels to the figure for site one and site two.

- It would be nice, if the structure of TUG-1460, which was used as a starting point for the development of TUG-2591, could be shown in the manuscript.

-A structure for TUG-1460 has been added to Supplementary Table 2.

- Discussion: The statement “...., to date it has not been possible to use NanoBRET to study binding to multiple sites on the same receptor, at the same time.” is not fully correct. In recently reported work (Huber et al. ChemMedChem, 2024, e202400284; Vogt et al. ACS Pharmacol Transl. Sci., 2024, 7, 1533-1545) NanoBRET based assays were used to detect binding to an intracellular allosteric binding site (by direct competition with an intracellular fluorescent ligand targeting this site) but also allowed to study

binding to distinct extracellular binding sites as a consequence of cooperative effects. Therefore, the statement mentioned above should be modified.

-Statement has been changed to indicate that current approaches do not allow direct detection of simultaneous ligand binding (Line 389). These references have been added.

- Reference list: For some references the page numbers (i.e., 16, 18, 30) are missing.

-References have been corrected

- Suppl Figure 1. What is the difference between the conditions TUG-2597 + T360 (orange triangle) and TUG-2597 + T360 (red rectangle)?

-This was a typo in the figure. The red square should be labelled as TUG-2597 + TUG-905, this has been corrected in the revised manuscript (Now Supp Figure 3).

- Suppl Figure 1 Aiii: The “.” in “TUG-2597 + T360. + TUG-905“ should be removed.

-Corrected

- Some synthesis schemes in the Supporting information (e.g., pages 15, 17, 41) are still shown in a “track changes mode” with lines at the left side and the bottom of the scheme. The same is true for the ¹H NMR of TUG-2598 (Figure S14)

-Corrected

- Supplementary References: For some journal names the abbreviation is used (e.g., Angew Chem Int Ed Engl) for other not (e.g., Journal of Medicinal Chemistry).

-Corrected

Reviewer #2 (Remarks to the Author):

Valentini et al. described three-color FRET, called “Dual Interaction Sequential Compound (DISCo-BRET).” A technique based on Bioluminescence Resonance Energy Transfer (BRET) was used to unravel the binding kinetics of two allosteric ligands to the Free Fatty Acid Receptor, FFA1. The authors spent time and effort synthesizing fluorescent compounds used in the assay. In my opinion, this is a method paper developed to study ligand binding to previously characterized binding sites to FFA1. Some effects are too small to convince and backed by enough evidence. Also, the manuscript

lacks proofreading because there are too many errors, mainly the ligand names. Below are some major and minor comments on the manuscript.

Major comments:

1. There is no supporting data provided in the manuscript that validates the luciferase (Nluc)-FFA1 fusions at both the N- and C- terminus are active (Ca²⁺ mobilization assay). While the generation of N-terminus fusion is referenced (Reference 16), the data demonstrating the functionality of the Nluc fusions of FFA1 compared to the unmodified (wild-type) FFA1 is absent in both the referenced article and the current manuscript. I highly recommend that the authors include this validation data in the current manuscript and direct readers to the relevant references. Without this, the claims regarding the functionality of the Nluc-FFA1 fusions remain unsubstantiated, and their similarity to the WT FFA1 is not confirmed.

-New data has been included (Figure 1D, Figure 2C, and Supp Figure 2) showing comparison of Ca²⁺ responses at N- and C- terminal Nluc tagged FFA1 as well as completely untagged receptor. Across these experiments it is clear that the pharmacology is broadly similar, with the only notable exception being that the site one agonists show somewhat higher efficacy in Ca²⁺ assays at the untagged receptor, while the site two agonists show slightly higher potency. Both of these observations are consistent with the untagged receptor expressing at somewhat higher levels than the tagged versions and do not suggest any inherent difference in receptor-ligand interaction.

2. In Figures 2 to 4, the competitive ligand TUG-2193 is mentioned. Similarly, another ligand named TUG-2456 is noted in Figure 4. However, these ligands were not explained in the text and the supplementary information, and no proper referencing or structures were also provided. Also, in the results section titled, 'Allosteric Regulation of FFA1 Ligand Dissociation' and the corresponding supplemental information, the TUG-2193 and T360 were interchangeably used in the figures, text, and supplemental information. It is unclear which of these compounds were used for the competition assay.

-We apologise for this error and the confusion. TUG-2193 is our internal compound name for T360, and TUG-2456 is our internal name for Cpd 6h. The manuscript and figures were originally prepared using our internal numbers, before switching to the published ones in the final version. A few instances were missed, which is why 2193 and 2456 were mentioned in some figures. We have fully corrected this and now in all instances TUG-2193 have been replaced with T360 and TUG-2456 replaced with Cpd 6h.

3. The choice of concentrations of compounds for the competition assays is not justified and well explained.

-Concentrations used in the competition assay were chosen based on: 1. using concentrations that were near the K_d, and 2. Using concentrations of fluorescent ligand that were unlikely to cause non-specific interference in the BRET assay. We deliberately

chose concentrations that were near K_d because DISCo-BRET by design can only measure binding when both tracers are present. Additional context has been provided in the results section explaining our choice (Line 230).

4. The Nluc in the C or N-terminus is shown to excite the terminal acceptor Sulfo-Cy5 along with the NBD dye, and the authors claim this energy transfer is not as effective as that of the NBD, as shown in Figure 3. However, in Figures 5F and 5G, when the authors use the single ligand NanoBRET format, the 'Specific BRET' between Sulfo Cy5/Nluc is much higher (~1 to 4) than that of NBD/Nluc (~0.01 to 0.04). The difference here is not explained well in the text, and the references to these figures in the Result section are wrong. Are both the green and red tracers still present in these experiments? If so, can the authors try an unlabeled site-one ligand as a control to verify this observation? For example, can the authors premix the green tracer (TUG-2597) with compound 26 in the synthesis steps of the red tracer (TUG-2591) before the competition assay with T360? In this way, one can directly observe the BRET values decreasing from the dissociation of the green tracer without the energy transfer to the red tracer and verify if the dissociation rate is similar to Figure 5F.

-Differences in the magnitude of BRET ratio relate more to the settings used on the plate reader than to true comparisons. Indeed, it is not typically possible to compare the BRET ratio for two different BRET pairs in this way. In this case we have used a wider bandpass to collect emissions for the red tracer than the green, but also the background luminesces will be much lower in the red wavelengths, so when expressed as a fold response above background the values will be higher. The point in relation to the magnitude of BRET in Figure 3 was possible because we have measured BRET by collecting emissions across the full spectrum.

The proposed experiment is interesting, but is complicated by the fact that adding the fluorophore to the compound will affect kinetics and may influence allosteric interactions. We therefore do not believe that this experimental design can be used to answer the question the reviewer notes.

5. In the Discussion section, lines 384 to 387, the authors mention that the ligand entry to site-one via the lipids impacts the binding of the site-two ligand. Can the authors explain the effect in more detail? Does the binding of the site-two ligand open or close the site-one? Overall, what is their hypothesis of the ligand binding at the site two effects on the site one based on their observations? Can the authors add a conclusion figure as a cartoon to summarize based on their binding kinetics?

-Text has been added to discussion to address this point (line 436) and a new summary cartoon Figure 7 has been added.

6. Labeling infra-red dye like Alexa750 or Cy7 would not overlap with Nluc emission, but overlapping only with NBD emission would significantly strengthen the conclusions made by the manuscript.

-In our experience it is difficult to find a dye that does not overlap at all with Nluc and the lower background observed with longer wavelength dyes means often poorly overlapping dyes actually work best in NanoBRET. There is clearly future scope to improve the assay with fluorophore selection, but in our view this would need to focus both on the spectral and chemical properties of the tracers. There are also practical considerations here that most BRET capable plate readers tend to only operate up to ~750 nm wavelengths, making an assay with longer wavelength dyes inaccessible to most researchers.

7. Data analysis was not explained adequately; it only mentioned that all data presented were carried out using Graphpad Prism 10 (GraphPad Software, USA).

-It is not clear what additional analysis the reviewer is requesting. Methods sections and figure legends explain how BRET data is being handled and shown, while in any place where non-linear curve fitting has been carried out details of which model used to fit the data is described in the figure legend.

Minor

comments:

1. In the Introduction, lines 65 to 66, the sentence is unclear, hard to understand, and must be rephrased. It also applies in the results section, lines 111-113.

-Both of these sections have been rephrased.

2. Does 'Cpd 6h' bind to sites one and two? This compound is listed in both the Supplementary Tables 1 and 2. Can this be explained?

-Cpd 6h binds to site two. It was originally included to show a reference full agonist response against the partial agonist site one compounds. For simplicity and clarity, we have removed this compound from Supplementary table 2.

3. Supplementary Table 2 is not referred to in the Results section 2, and Supplementary Table 1 is referred to instead.

-Corrected

4. In Figure 2, which details the development of the red tracer to orthosteric site one, the Ca²⁺ mobilization data were normalized with the binding of T360. However, it is mentioned as a site two ligand. Can the authors clarify in the text why they chose T360?

-T360 was used because it was important to show that the compounds expected to bind to site 1 are partial agonists, this is easier to follow if data are normalized to a full agonist. Clarification has been added to the results text (Line 166).

5. Line 240-241; "In contrast, the site two compounds, T360 and Cpd 6h, display clear competition in both DISCo-BRET and green tracer binding (Fig. 4Aii and 4Bii. I didn't see any of these compounds were tested here. Also, there is no effect of aLA, but the authors mentioned it as a modest effect.

-As noted in our response above to reviewer 1, TUG-2193 and TUG-2456 are our internal names for T360 and Cpd 6h respectively. We have corrected the compound names in the figure. Although aLA does not fit a full curve, it is clear that it is starting to have an effect at the highest concentrations tested, text has been modified to clarify this (line 246).

6. In Figures 3C & 3D, the same symbol has been used to represent the Specific BRET values for the conditions 100 nM TUG-2591 and control (TUG-905/TUG-2193). It needs to be fixed.

-Corrected

7. In Figure 5, it is hard to distinguish the blue and green curves. I recommend that the authors try different symbols for them.

-Colours have been changed.

8. The caption for Figure 2B and the corresponding Results sections detail using the ligand Cpd 6h for Ca²⁺ mobilization assay. However, the Cpd 6h data is not shown or mislabeled in Figure 2B.

-This was an error in the figure legend, it should have listed T360 as the site 2 compound. In our original submission it was labelled with our internal TUG number. These issues have been corrected in the revised manuscript.

Reviewer #3 (Remarks to the Author):

In the manuscript entitled “Multi-coloured sequential Resonance Energy Transfer for simultaneous ligand binding at G protein-coupled receptor” written by Valentini and collaborators, the authors aimed at investigating the interaction between the binding site of various ligands and more specifically the interactions between different allosteric binding sites. The authors developed an original technical approach based on sequential RET between a receptor (Free Fatty acid Receptor (FFA1)) and fluorescent ligands labeled with NBD and SulfoCy5 and specific for the allosteric binding site 1 or 2.

General consideration
As mentioned above, the approach is original and, whereas the various attempts at sequential RET strategies developed to study receptor oligomerization, for example, are not convincing, the demonstration of sequential RET is here rather well done. But the overall impression is that the manuscript suffers from numerous inconsistencies and approximations: the panels in the various figures do not correspond to the description given in the manuscript, some ligands do not have the same name in the text

and in the figures, references to the figures in the text are not well done and the reader has to find the right panel corresponding to the text.

Thank you for your identifying these issues. As noted in our response to comments from other reviewers, for a few of the compounds (T360 and Cpd 6h) our internal reference numbers had not been replaced with the published compound names in several of the figures. We have now carefully reviewed all figures, text and figure legends to address this issue. We have also carefully proofread the manuscript to ensure all figure panel references are correctly called out in the results text.

Values for ligand affinity, K_{off} and K_{on} are given without error (SD or sem) except in Suppl. Table 3.

-We have updated all tables showing ligand binding kinetic data to include 95% CI ranges for all binding constants that have been experimentally calculated.

Finally, the conclusions drawn from the results are not discussed.

-Key conclusions are highlighted in the results section, and additional discussion added to the discussion section.

From a scientific point of view, as mentioned above, the strategy developed may be interesting, but it doesn't allow to consider interactions between sites. It is simpler to perform two BRET measurements with each of the tracers. The authors mention that the identification of allosteric modulators is a major focus of GPCR research, but Disco-BRET is too complicated to finely characterize new molecules. Finally, the benefits of DISCo-BRET compared with Nano-BRET are unclear. It seems much easier to perform and interpret Nano-BRET experiments with the Nluc-FFA1 receptor and each of the two tracers separately than to perform Disco-BRET.

-DISCo-BRET allows for direct detection of binding of two ligands to the same receptor at the same time. Indeed, it is true that allosteric modulation of affinity can be studied using other approaches, but these typically require saturating concentrations of modulators to be sure measurements are from receptor populations bound to both ligands. DISCo-BRET allows for this to be assessed at submaximal concentrations, as the signal already only comes from receptor with both ligands bound. In the current manuscript we show the approach can help characterise novel pharmacology, quickly identify binding sites, and show effects on binding kinetics. We are also confident that future studies will identify more important uses for this novel approach.

Indeed major questions have to be answered before using such a technique, especially to investigate the interactions between binding sites:

- The proportion of receptors labeled with each of the two tracers and with the two tracers simultaneously: in the present study the concentrations of the tracers are not compatible with a saturation of the two binding sites. Therefore, the authors have to deal with a mixture of the different complexes and the values for the affinity, K_{off} and K_{on} correspond to the mean of the values of the different species.

- While it is true that the concentrations used in our initial studies were subsaturating, it will not be an average of the mixture of complexes, given that DISCo-BRET only measures the specific complex bound both ligands. To highlight this point, we have now added additional kinetic and saturation binding studies using much higher concentrations of the tracer ligands.

- In the disco-BRET configuration, the red signal and the green signal are both dependent on the binding of both tracers. A decrease of the occupancy of site 2 can induce a decrease of the red signal but it is not necessarily due to a decrease of the occupancy of site 1. Conversely, a decrease of the occupancy of site 1 can induce an increase of the green signal without any increase of the occupancy of site 2. It means that it is almost impossible to correlate the variation of the signal with the occupancy of the binding sites.

-It is true that in DISCo-BRET the signal for the red tracer is completely dependent on occupancy of the green tracer. This is the fundamental goal of the approach. Indeed decrease in occupancy of the green tracer (as is seen when competing with site two ligands), decreases DISCo-BRET signal. The second point where decreased occupancy of site 1 can induce an increase in green signal without occupancy of site two is partially true, as decreased FRET between the ligands could lead to some increase in site two signal, but mainly we see increase in site two signal when different ligands bind at site one that either modulate site 2 affinity, or alter receptor conformation. However, we disagree with the comment that it becomes impossible to attribute changes in DISCo-BRET to specific binding, as we have effectively done this by carefully using competing ligands for the two sites independently or in combination.

- The addition of a third ligand as a potential competitor leads to an even more complicated situation. Since positive or negative cooperativity are dependent on the ligands themselves, the final result of one experiment is dependent on the cooperativity between the tracers and between the unlabeled ligand with the tracer. For example, the addition of TUG-905 to displace TUG-2591 modifies the cooperativity between TUG-2591 and TUG-2597, and that was what the authors were looking for, but the binding of TUG-905 introduces a new type of cooperativity between TUG-905 and TUG-2597. Therefore, the variations of the green signal are dependent on both cooperativities which are not necessarily the same.

-We agree there is complexity here, particularly when trying to assess allosteric modulation of site two by competing site one ligands in DISCo-BRET competitions. This is why we have not aimed to explicitly quantify this cooperativity in this study. We have simply highlighted that there is enhanced green tracer signal when competing ligands are added, suggesting that this could be due to increased affinity and/or conformational changes. However, the primary read out of DISCo-BRET is the emission from red tracer, which as we show can be interpreted based on which binding site is being competed with. Taking this further, we have added new competition studies with novel FFA1 antagonists, and shown that the assay can be used to quickly identify likely allosteric interactions between sites, which we then confirm in functional Ca²⁺ antagonism studies.

- BRET signal variations in the manuscript have always been associated to a variation of the occupancy of the binding sites but the binding of a ligand, especially agonists as done in the present study, can have a deep impact on the structure of the receptor. This leads to variations in the distance between the Nluc and the tracers and therefore to variations of the BRET signals. This point has never been discussed or even considered in the manuscript.

-We thank the reviewer for this insight suggestion. We have now added discussion to this point through the manuscript. In particular, we note our new data in Sup Figure 9 where we have used near saturating concentrations of the tracer ligands, does suggest that part of the increase in BRET between the C terminal Nluc and our Green site 2 tracer that occurs following addition of site 1 competing ligand (TUG-905), must be related to such a conformational change. This is discussed in results (Line 324).

- The authors have used fluorescent agonists. Did they evaluate the potential consequence of the binding of the tracer on receptor internalization?

-We agree this is an important consideration when using agonists over long time courses in intact cell assays. However, we note that FFA1 has not traditionally been associated with strong internalisation. Even so, to determine if internalisation could be affecting our results we have now added bystander BRET internalisation assays measuring movement of C terminally Nluc tagged receptors away from a CAAX membrane anchored mNeonGreen fluorescent protein (New Sup Figure 8). These data show that although the assay can easily detect internalisation of GPCRs (shown using the β 2-AR-Nluc), we do not observe any internalisation of FFA1-Nluc when treating either with our competing ligands (TUG-905 or T360), or with our tracer ligands (TUG-2591 or TUG-2597). In fact, both TUG-905 and T360 have a tendency to increase cell surface expression. Description of these studies has been incorporated into the results section (Line 301).

- The discussion on the reciprocity between the binding sites is not accurate.

-Removed

- Finally, the authors should have discussed the interest of the strategy for other GPCRs. It is most probable that the DISCo-BRET can be relevant only for receptors for which a binding site equivalent to the binding site 2 (i.e. located deep inside the receptor) has been described.

-Additional discussion has been added to address the extension of DISCo-BRET to other GPCRs (Lines 415-424).

The authors will find below different points which can refer to the points mentioned above.

Specific points

-1- The manuscript is not easy to read and suffers from inconsistency. Throughout the manuscript, the authors mentioned the ligands T360 and Cpd 6h and referred to different figures and tables. But none of these two ligands were mentioned in the figures. For example, lines 240-243: “In contrast, the site two compounds, T360 and Cpd 6h, display clear competition in both DISCo-BRET and green tracer binding (Fig. 4Aii and 4Bii). It is notable that these compounds did not result in complete competition of the DISCo-BRET (Fig 4. Aii; Sup. Table 2)”. We ended up assuming that T360 and Cpd 6h correspond to TUG 2193 and TUG 2456 but without being completely sure.

-As noted above, this was an issue in compound naming which has now been corrected.

-2- In the introduction, the authors wrote “... with studies demonstrating at least two distinct allosteric sites on the receptor” (line 87). It is not clear why these binding sites have been called allosteric and not orthosteric? Moreover, usually allosteric ligands do not promote receptor activation (except the ago allosteric ligand) but, in the present study, the authors have selected the green and red tracers for their capabilities to activate fully the receptor.

-Ligand binding site pharmacology around FFA1 is complex and cannot be fully discussed here given word limits. Both sites have been described historically as allosteric due to the fact that ligands binding to each site have been shown to modulate activity of fatty acid ligands. As the reviewer notes, traditionally allosteric ligands are thought of as modulators that do not activate the receptor directly, but in the case of FFA1, all of these ligands behave both as modulators and direct agonists. Further complicating FFA1 pharmacology, is the fact that ligand binding to both sites are essentially synthetic fatty acids, and therefore it has been unclear if the fatty acid endogenous ligands bind to a different site all together, or to one or even both of the sites that have been described as “allosteric”. More recent structure work has suggested fatty acids can bind to “site 1”, suggesting that it is in fact orthosteric, but this still does not explain the fact that fatty acids show functional cooperativity with other site one agonists. Due to this complexity, we have chosen to simply refer to the sites as site 1 and site 2 through the manuscript, as opposed to referring to them either orthosteric or allosteric.

-3- Most of the values of the potency or affinity are indicated without errors (example line 130, Figure 1E or F). Please provide SD for all the values. Also indicate for all the graphs if the plots correspond to pooled data from independent experiments or to a “representative” experiment.

-All values for potency (EMax), affinity (pKi) and efficacy (EMax) do indeed include calculated SEM values in the supplementary tables. For all calculated on and off rate constants we have now also added 95% confidence intervals. All figures already indicated that data are shown from N=XX experiments. All data are pooled and in no cases have data been shown from “representative” experiments. A statement to this effect has been added to the online methods (Line 601).

-4- line 184 : “ ... that incorporating physical separation ...” can give the idea of a physical barrier. “ increasing the distance between Nluc and the red tracer, TUG2591, ...“ is maybe better.

-This change has been made as suggested

-5- Figure 3: it would have been better to specify on all the panels what “specific BRET” means, as done in Figure 4.

-In Figure 4 and Figure 3E/F specific BRET is qualified based on the ratio that has been measured. In the other panels of Figure 3 (C/D) BRET data is being shown across the full spectrum of wavelengths, it is therefore not possible to include a comparable description in these panels. The figure legend for C/D clearly indicates that specific BRET is the BRET signal measured above the TUG-905/T360 treated condition.

-6- Why are the legends of Y-axis for the Disco-Bret in Figure 3 E and F (Disco-BRET (700nm/525nm)) and in Suppl Figure 1 (Specific BRET 690/540) not the same? Is there a difference in the ratio? If so, why?

-As noted in the manuscript, one experiment has taken the ratio from the area under the spectra scan curves while the other was taken from discrete measurements filtering for the indicated wavelengths. Long bandpasses are used for these measurements, and ultimately slightly different middle wavelengths were used for the individual measurements vs the area under the spectra scan.

-7- Are the legends of suppl Figure 1 Aiii and Biii correct? It seems that TUG-2597 + T360 corresponds to two symbols but TUG-2597 + TUG-905 is not present.

-This error has been corrected in what is now Sup Fig 3.

-8- There is no mention of T360 and Cpd 6h in Figure 4 and no mention of TUG 2193 and TUG 2456 in the text of the manuscript. Therefore, it is difficult to follow the argumentation.

-As noted above, TUG-2193 is our internal reference number for T360, and TUG-2456 is our reference number for Cpd 6h. We apologise for this error and have corrected the manuscript to only refer to T360 and Cpd 6h.

-9- In the description of the results illustrated in Figure 4 Ai and Bi, there is no comment on the variation of the slope of the curves. In panel Bi, the slope of the curve can be greater than 1 (TUG-905) and lesser than 1 (TUG-770 and Cpd-B) suggesting potential positive or negative cooperative interactions between the binding sites 1 and 2, depending on the competitor. Because in Panel Ai, the BRET signal corresponds at the same time to the presence of the red tracer but also the green tracer, one would expect to see variation in the slopes in the competition curves, as in Panel Bi. It is not the case. The authors have to comment such discrepancies. Moreover, it is mentioned in the “Data Analysis” section that data were fitted to a three-parameter log agonist concentration vs response model. It is obviously not the case in Figure 4 Bi.

-Data in panel 4Bi had been fit to a two site fit logIC50 model, instead of a one site model in error, which is why the slopes appear different for different ligands. While we agree that slopes may vary due to cooperativity, the fact that we see little cooperativity between the two tracer ligands and broadly similar cooperativity for each site one ligand in relation to its effect on the Green tracer, we do not think it is unexpected that the slopes are all broadly the same.

-10- lines 250-251: “that both T360 and 251 Cpd 6h actually have positive cooperativity with the red tracer (Fig. 4Ci; Sup. Table 3)”. It could be the case but one can also imagine that the increase in the BRET signal induced by two compounds in Panel Ci is due to a structural modification of the receptor leading to shorter distance between Nluc and the red tracer. The authors have to discuss that point, even if it is less probable.

-We thank the reviewer for this insightful comment and completely agree that some aspect of this increase in BRET may (and indeed almost certainly is) due to conformational changes. This point has been mentioned in the results section related to Figure 4 (Lines 242-244), and addressed again in discussing our new kinetic binding data employing near saturating concentrations of the tracer ligands (New Sup Fig 9). The fact that TUG-905 still increases NBD/Nluc BRET, even with near saturating concentrations of the tracers does indeed suggest that part of this increase is due to conformation change. Further supporting this point is the fact that the kinetics of TUG-905 response in these new studies shows that the red tracer must first dissociate before TUG-905 can have the effect, again consistent with the expectation that the receptor pools will have been at essentially saturating levels of both tracers in these new studies.

-11- The authors developed a sequential RET system to study the interactions between the binding sites 1 and 2. But one of the stronger evidence of interactions between the binding sites is provided by Figure 4 Ci and Cii when using the Nluc-FFA1 receptor and measuring the BRET with the two tracers independently. Of course, it needs two experiments but the interpretation is much more reliable. Therefore, is it necessary to develop a sequential RET assay to study binding site interactions or are the investigations on the interactions between the binding sites the best model to prove the interest of Disco-BRET?

-DISCo-BRET is a new and innovative approach to study ligand binding to GPCRs. It is true that it is more complex, but it also allows for the study of multiple binding sites at the same time in a single experiment. Further, because DISCo-BRET allows us to specifically measure binding from receptors that have both ligands bound simultaneously, it allows us to study binding and allosteric interactions while using lower, submaximal concentrations of the tracer ligands (See responses to reviewer 4 below). We have demonstrated several ways DISCo-BRET can be useful, including studying binding kinetics and quickly identifying novel antagonists with unique pharmacology (See New data added in Fig 4 Aiii, Biii, D and Sup Fig 7). We have already demonstrated these potential uses and we fully expect that future work will identify further new and innovative ways to use DISCo-BRET to study GPCR ligand binding.

-12- lines 264-265: “Tracer binding was monitored as either DISCo-BRET to assess binding of the red tracer (Fig. 5B)”. The formulation of the text is not correct since Disco-BRET is dependent on the binding of both tracers. ““Tracer binding was monitored as either DISCo-BRET to assess simultaneous binding of the green and red tracers (Fig. 5B)” would be more accurate.

Moreover, in Figure 5, TUG 2591 and TUG 2597 were used at 100 nM and 316 nM, respectively. At these concentrations, when considering the affinity reported in Table 4, respectively 250 and 110 nM, we can estimate that about 30% of site 1 and 75% of site 2 are labeled. It means that some of the receptors are empty, others bound to one of the two ligands and a small proportion to the two ligands (~23%, if we do not consider any cooperativity). Therefore, it means that when using the Disco-Bret (BRET 690/540) and the nanoBRET (540/450), the authors dealt with different types of receptors depending on whether they are linked to one (TUG 2591 or TUG 2597) or two ligands (TUG 2591 and TUG 2597). In these conditions, it is difficult to compare the signals.

-Thank you for the suggestion. We have made the recommended change to the results text. We have also repeated the experiments presented in Figure 5, now using 10x higher concentrations of both tracers. While it is true that the concentrations we have used originally will mean that receptor populations will be mixed in terms of binding one, the other or both tracers, the strength of DISCo-BRET is that it will still only measure the response from those receptors with both ligands bound. To demonstrate this, we have repeated the experiments from Figure 5, now using 10x higher concentrations of each tracer (New Sup Fig 9, Data also added to Sup Table 4). From these data it is clear that the

off rates from DISCo-BRET do not change with the different concentrations of tracer. It is however noted that the effect TUG-905 has enhancing green tracer BRET does change (Sup Fig 9E). This is interpreted as at submaximal concentrations of red tracer TUG-905 will quickly bind to receptors that do not have red tracer bound, enhancing binding (or BRET through conformational change) of the green tracer. In contrast, when near saturating concentrations of red tracer are used, TUG-905 can only bind after the red tracer has dissociated. This point is clearly apparent when observed that the rate of increase in BRET following TUG-905 addition in Sup Fig 9B (monitoring NBD/Nluc) appears to perfectly match the rate of decrease in BRET in Sup Figure 9A (Monitoring SulfCy5/NBD). Description and interpretation of these new experiments have been added to results (Lines 312-327).

-13- lines 268-270: "It was notable that the off-rate for the red TUG-2591 from site one was substantially slower than that of the green TUG-2597 from site two." Please indicate the figure to be referred to. Moreover, if we consider values reported in Suppl. Table 4, it is not obvious, the k_{off} are respectively 0.41 and 0.38 for TUG-2591 and TUG-2597, respectively. Finally, in Figure D, the decrease of the signal can either be due to the dissociation of the red or the green tracer.

-The data referred to in line 268-270 is describing the data from DISCo-BRET binding studies, showing an off rate of 0.036 - 0.032 (depending on which tracer concentration was used) for TUG-2591 compared with 0.18-0.27 for TUG-2597. The results text has been modified to make it more clear which data was being referred to (Line 295-297).

-14- Figures 5 F and G do not correspond to the legend beneath. It seems that panels have been inverted. And the different panels referred to TUG 2193 while the legend and the manuscript referred to T360. All these approximations make the reading of the manuscript very difficult! Moreover, it would have been very interesting to use the same protocol used in Panel F and G and to follow the dissociation of TUG -2591 and TUG 2597 in the presence of an excess of T360 and TUG-905, respectively.

-This figure legend and figure callouts in the text have been corrected. We agree it would be potentially interesting to see how T360 and TUG-905 affect dissociation in single ligand experiments, but we do not believe these studies provide a meaningful addition to the current study, given that the way these ligands will modulate TUG-2591 and TUG-2597 respectively will not be the same as the way TUG-2591 and TUG-2597 modulate each other.

-15- Lines 293-296: "When assessing site two ligand binding, association was observed after addition of green TUG-2597 alone, which was enhanced by the presence of TUG-905, consistent with positive cooperativity between these ligands." It is not clear to which panel the text referred, please indicate the panel to be considered. If it is Panel C, the conclusion is hasty. Various hypotheses can be formulated. First, nothing indicates an increase of the association but simply an increase of the signal. Because the red tracer is displaced by TUG-905, there is no more FRET from the green tracer to the red tracer

and therefore the fluorescence of NBD increased. Second, there is no proof of cooperativity.

Moreover, and by contrast to what panel A and D can suggest, the proportion of complexes receptor/green tracer/red tracer is not known since the concentration of the tracers used do not allow a saturation of the binding sites. This is a major default of the study.

-An appropriate figure callout has been added. We have also qualified the text to say that the “signal” was enhanced, not the binding (Line 339).

-16- lines 308-310: “As before, binding of green tracer was measured using the NBD/Nluc BRET ratio (Fig. 6F), while binding of the red tracer was assessed through DISCo-BRET and SulfoCy5/NBD ratio (Fig. 6E).” Again, panels 6E and 6F as described in the text do not correspond to the panels in Figure 6 ... probably another inversion!

-Corrected in the results text.

-17- lines 313-314: “These results demonstrated a rapid binding of green tracer, which was not impacted by the addition of red tracer, although green tracer binding was enhanced by pre-treatment with the site one competing compound, TUG-905 (Fig. 6E). Addition of red tracer did however lead to a clear increase in DISCo-BRET, but notably this occurred with a relatively slow association rate (Fig. 6F; Supp. Table 5).” If the complex receptor/green tracer/red tracer is one of the major species in the preparation, the addition of red tracer should necessarily impact the NBD/Nluc signal. The absence of variation could reflect a low proportion of receptor/green tracer/ red tracer complex.

-We agree that the lack of decrease in NBD/Nluc BRET upon addition of the red tracer does indeed suggest a relatively low proportion of receptors with both ligands bound. Again, the nature of DISCo-BRET is that it allows us to measure red tracer binding specifically from this pool of receptors. A clarification in the results text has been added (Line 360).

Requested experiment:
1) saturation binding curves (equivalent to Suppl Fig. 1) and kinetics curves (equivalent to Fig. 5D/E) for TUG-2591 (red) in DISCo-BRET mode, with for each a plateau at high concentration

-We have repeated saturation binding experiments shown in Figure 3/New Sup Fig 3 using 10x higher concentrations of Green tracer, TUG-2597. We have also increased the concentration of red tracer used to more fully capture the saturation binding response. Data from these experiments are added to the results (Lines 219-223). Importantly, the data show the affinity for TUG-2591 measured through DISCo-BRET is comparable regardless of the concentration of TUG-2591 used (sub maximal, or saturating). This again highlights a central advantage of DISCo-BRET that it measures red ligand binding affinity only from the pool of receptors with green ligand bound.

2) The effects of the competitors have to be investigated on both binding sites in NanoBRET when using Nluc-FFA1 receptor. The authors have to be sure that receptor occupancy is high enough to favour one complex species (at least occupancy of more than 75% of the binding site occupancy, i.e. at least 3x Kd) : i) TUG-2597 (green) 316 nM alone, followed by TUG-905 high concentration to clearly establish TUG-905 effect on TUG-2597 ; ii) TUG-2591 (red) 750 nM alone, followed by T360 high concentration to clearly establish T360 effect on TUG-2591

-We agree that these studies are required to establish allosteric modulation, however defining modulation of TUG-2597 and TUG-905 is not a central objective of this work. We show that site one compounds do enhance binding signal for TUG-2597 and report only “pIC50” values, without making any direct attempt to quantify allosterism. We have also now added additional context to highlight that some aspect of the change in signal could be due to conformation change in addition to effects on binding affinity

3) Do Fig. 5 B/C/E/F experiments again with similar receptor occupancies for each tracer, preferably with ~90% occupancy (concentration = 9 x Kd).

-These experiments have been repeated using 10x higher concentrations of both tracer ligands, the data from which are now showing in Supp Figure 9. As described in new results text addressing these experiments (Line 312-329), increasing the concentrations of tracers had no effect on the dissociation rates measured for TUG-2591 or TUG-2597. It was however notable that the enhancement of TUG-2597 signal following treatment with TUG-905 was affected.

Conclusion : we consider that the manuscript can be published in its current state.

Reviewer #4 (Remarks to the Author):

The manuscript by Valentini et al., “Multi-Coloured Sequential Resonance Energy Transfer for Simultaneous Ligand Binding at G Protein-Coupled Receptors,” is a very interesting study where an attempt is made to observe the simultaneous binding of several ligands to the receptor. The topic addressed in the publication is very important to the field – novel ligand binding assays to G protein-coupled receptors are needed to gain further insight into the pharmacology of these proteins. The focus of this publication is the NanoBRET assay, which has gained momentum in recent years and has become increasingly popular among the GPCR community. In that regard, the reference list should reflect the publications of different workgroups working on the subject. The developed DISCo-BRET assay is a promising approach for performing multiligand binding

study. However, the manuscript in its present form raises several considerations which need to be addressed before it is ready for publication:

-Additional references highlighting key advancements in NanoBRET have been added.

Fundamental:

1. The proposed method and approach are very elegant and interesting. However, avoiding bleed-through to red fluorophores from the Nluc energy donor is very challenging, especially when multiple fluorophores are involved. Although "physical separation" has been mentioned, no data are available to rule out potential optical interference. The description of physical separation described (lines 187-194) has not been confirmed by experimental data.

-An additional reference (Hughes et al. 2014) has been included describing characterisation of SulfoCy5 fluorophore chemical properties.

2. If FRET between ligands is indeed occurring, it should also be observed in wild-type receptors when the corresponding light or laser activates the green fluorophore. This would also exclude the possible effect of Nluc on the properties of the receptor. This kind of control experiment would give additional proof of multiple ligand binding.

-We thank the reviewer for this useful comment and agree that this conceptually would be possible. However, our NBD tracer has been optimised for BRET based approaches, and indeed is not generally suitable for imaging/microscopy, making it very difficult to complete FRET studies. This is particularly true given the lipophilicity of FFA1 ligands, meaning that these ligands tend to show significant non-specific membrane binding, hence why using BRET to measure their binding has been critical.

3. It is unlikely that the classification of FFP1 allosteric ligands by efficacy is as straightforward as described in the publication ("Since site two FFA1 ligands behave as full agonists in Gαq signaling, while site one ligands are partial agonists..."). It is widely known that the classification of compounds as full or partial agonists is highly dependent on the assay, expression levels and other used parameters making the results inconsistent. The publication gives the impression that site two targeting ligands cannot be partial agonists.

-We completely agree with the reviewer that full vs partial agonism in a Ca²⁺ assay is an oversimplification of binding site classification. However, it is broadly effective for FFA1. That said, to extend this, we have also classified key ligands in an arrestin recruitment assay (New Sup Figure 1). Interestingly, these data show that Site 1 agonists are full agonist of arrestin and site 2 agonists are partial (exactly the opposite of what is seen in Ca²⁺). The trend is consistent for our tracer ligands, and these data have been used to further confirm the likely binding site of the tracers. To our knowledge, this is the first time site 2 FFA1 ligands have been shown to be partial agonists in arrestin.

Major:

4. The abstract contains minimal information. There is no indication of the receptor that has been studied nor is the essence of the method described.

-Reference to FFA1 has been added to the abstract. The abstract does clearly state that our approach is based on sequential resonance energy transfer between two fluorescent tracers. Given word limits for the journal, it is difficult to add much more detail.

5. The binding and potency parameters in the text and kinetic parameters in the Supplementary Tables are given without confidence intervals. It makes it difficult to evaluate the significance of the obtained results and the correctness of the conclusions. Also, it is not written what the error bars on the figures correspond to (SD or SEM?) and whether the data are pooled or from single experiments.

-Potency, affinity and efficacy values all have SEM shown in their supplemental tables. We have not included these values in the main results section to avoid overcomplicating the text. We have also now added 95% CI for calculated kinetic parameters to the supplemental data tables. It is stated in the data analysis part of the method section that all data are shown \pm SEM and we have modified this to also clarify that data are shown "pooled data from the indicated number" of independent experiments (Line 602).

6. T360 and Cpd 6h have been used as site two ligands (lines 240 -), but no data about these ligands have been presented in Fig. 4. Instead, the figure has data of ligands not described in the text. Fig. 2B text also indicates that the graph should have data of Cpd 6h, but there is no corresponding curve.

-As noted to the other reviewers, TUG-2193 and TUG-2456 are our internal reference numbers for T360 and Cpd 6h respectively. Unfortunately, in our initial submission there were several cases where the TUG numbers were still used in figures. These have all been corrected, and in all cases T360 and Cpd 6h are now used.

7. The dissociation rate constant depends on the extent of the dissociation (span). As it is not the same for all processes (Fig. 5), it is important to present these data and consider them while interpreting results.

-It is not fully clear to us what the reviewer is asking for here. We have added additional dissociation experiments using higher concentrations of both ligands. Data are all fit to the indicated ligand dissociation binding models.

8. The developed assay system could not detect the effect of the antagonist GW1100 on the fluorescent ligand binding. To confirm that GW1100 interacts with this receptor and behaves as an antagonist, experiments with Ca^{2+} mobilization would have to be performed. Only then can we make some conclusions about the localization of the antagonist binding site.

-We have now added Ca^{2+} experiments showing that GW1100 does functionally inhibit both a site one ligand, TUG-905 (Sup Fig 6A) and a site two ligand, T360 (Sup Fig 6B). Appropriate results text has been added (Line 259-260).

9. Fig. 2 F and G indicate substantial BRET connected with the binding of purple ligands to C-terminal Nluc. The presentation of these binding parameters would be important for the interpretation of the obtained results.

-It is not fully clear what the reviewer means by “purple ligands”, but I think they are referring to the data shown in these graphs with purple symbols. Indeed, there is a small amount of direct BRET that remains from the C terminal Nluc to TUG-2591, and this has been considered in discussion of later data with this compound (ex line 250-255).

10. The modulation of fluorescent ligands' binding was also studied in the presence of the site one ligand TUG-905 (Fig. 6), but there is no information on how the pretreatment with site two ligands would alter the binding.

-Additional context has been included (line 335-336).

11. There are indications that some data is excluded from the graphs. For example, Fig. 2D is missing value of -10, Fig. 2F has eight datapoints, while Fig 2G-H have seven, Fig 3F has no 17.8 nM datapoint. The readers would appreciate all the measured data to be displayed on figures.

-No data points have been excluded from the results. Due to practical considerations of experimental design there were some cases where different numbers of points were used on concentration or saturation binding data. The data from Figure 3F does not have 17.8nM because this concentration was not used in the FFA1-Nluc assay. It was however incorrectly included in the legend of the data in 3D (although there were no actual data for it on the graph). We have also added new data the new Sup Figure 3D that includes a more broad range of concentrations (including 17.8nM) of TUG-2591 in a DISCO-BRET saturation binding assay.

12. Were the association curves on Fig. 6B-C, E-F measured longer? The curves look like the association is not complete.

-The full data set is shown. The fact binding association was not complete is why longer time courses were used for the follow up experiments shown in 6G/H.

13. Is 10 μM concentration of unlabeled ligand T360 ($\text{pIC}_{50} = 7.1-7.4$) enough to saturate nonspecific binding in the presence of 3 μM TUG-2597? Usually, great excess of an unlabeled ligand is used to determine non-specific binding.

-The concentrations were selected based on the effectiveness of these compounds to compete in competition binding data (Figure 4). In all kinetic data shown in our original submission, TUG-2597 was used at 0.316 μM not 3 μM . In the revised submission we have added additional data where we have used 3.16 μM TUG-2597, but in these studies 30 μM T360 was used to set NS binding.

Minor:

14. Data in Fig. 1E and F repeat (in normalized form) data presented in Fig. 1G and is an unnecessary duplication.

-We disagree that the data in G is not useful given that it highlights the difference in BRET signal obtained when using the N vs C terminal construct, which cannot be assessed from the normalized data in E and F.

15. Fig. 4 would be easier to follow if the ligands are depicted in unique colours and the TUG-905 reference curve would be included with the datapoints.

-We are not sure what the reviewer is suggesting here. Given the number of competing ligands, particularly now with the addition of more antagonists, it would not be possible for each to have its own unique colour. We prefer showing only the curve for repeat of TUG-905 data to emphasize that it is included only for reference and is not new data.

16. Some conclusions about the efficacy of compounds are slightly misleading. For example, Fig. 1B compound TUG-2490 does not reach the same level of activation as other full agonists

-We agree that the data present do not fit full curve with well defined E_{Max} values. While the estimated curve fit values (Sup Table 1) do suggest that they are likely full agonists, we have qualified the statement in the results text to indicate that true efficacy can not be determined.

17. There is no Fig. 5H (line 280).

-Corrected

Reviewer #2 (Remarks to the Author):

In response to comment #6, the authors discuss potential improvements in Nano-BRET assays and plate readers to accommodate far-red dyes. I recommend incorporating this information into the third or final paragraph of the discussion section.

-An addition sentence has been added to the discussion to address this (line 424-426).

Minor comment:

In reference ID 25, there is an error displaying "INVALID CITATION."

-Corrected

Reviewer #3 (Remarks to the Author):

Major points

-1- The absence of reciprocity: the green tracer has an impact on the red tracer but, conversely, the red tracer has no impact on the green tracer link. This phenomenon of non-reciprocity was addressed in the first version. Our comment in the previous review may have suggested that this aspect was not important. Perhaps we worded it wrongly, but it is an important aspect, but the explanations provided in the first version did not seem convincing to us.

How the authors explained this absence of reciprocity? They have to discuss this point in depth.

-A more extensive discussion of allosteric reciprocity has now been included (Line 461-473). Several possible explanations are considered, including that the binding affinity is not affected, or that the full agonist is stabilising a G protein bound state, which in turn affects binding kinetics at site one.

-2- The authors never considered the fact that the emission of TUG-2597 should change depending on whether or not there is an acceptor (TUG-2591) nearby. Indeed, it should decrease when TUG-2591 was added (Figure 6E) and the competition with unlabelled competitors for TUG-2591 (TUG-905 for example) should induce an increase of the emission (Figure 4Bi).

We asked the authors to discuss that point and not to consider only "either cooperativity or a conformational change resulting in enhanced BRET between the C terminal Nluc and the green tracer". (line 242-244). The authors should discuss why displacement by unlabelled competitors of TUG-2591 bound to the receptor and close to the donor (TUG-2597) is not a plausible hypothesis to explain the increase in the specific BRET ratio. They also should explain why the addition of TUG-2591 in Figure 6E did not induced a specific BRET ratio decrease.

To our opinion, taking into account the % of receptors occupied concomitantly by TUG-2591 and TUG-2597 should provide part of the answer

-Additional statements have been added to address this point, including highlight sequential energy transfer as a possible cause for the increase in NBD/Nluc BRET (Line 244-245). This point is also now discussed in relation to the single ligand Nano-BRET competition studies in Fig4Cii, which show that similar increases BRET ratio are observed even without the red tracer present, therefore indicated that this change in sequential transfer is not the primary cause of the increased NBD/Nluc ratio (Line 291-283). Finally, as requested additional discussion around data presented in 6E is included to highlight that the reason addition of red tracer here does not lead to decrease NBD/Nluc signal, is likely because the experiments have been carried out at subsaturating concentrations of the tracers (Line 374-377).

-3- Validation of the model of ligand binding: to validate the model, we ask the authors additional experiments in BRET and compare the results to those already obtained in DISCO-BRET:

1) BRET Kinetics of saturation on Nluc-FFA1 with the same format than the one of Suppl 9A

- Labelling with TUG-2591 then addition of T360 and later addition of TUG-905(dissociation)

- Labelling with TUG-2597 then addition of TUG-905 and later addition of T360 (dissociation)

-These data have now been added in Sup Figure 9G, 9H, 9J and 9K, as well as curve fit values in Sup Table 4.

2)Kinetics of saturation on FFA1-Nluc: labelling with TUG-2597 and addition of TUG-905 and later addition of T360. This experiment is going to be useful to determine the effect of non-fluorescent site 1 agonists on NBD/Nluc ratio using the construct used for Disco-BRET and to discuss the possible conformational change, and its effect on non-visible allosteric reciprocity.

-These data have now been added in Sup Figure 9F and 9I, as well as curve fit values in Sup Table 4. Broadly the data show that there is a clear immediate increase in BRET when adding a saturating concentration of TUG-905, this occurs with either the N or C terminal construct. The data suggest the effect is largely due to conformational change. Dissociation rates have been calculated for the tracers in the presence of saturating concentrations of unlabelled ligands binding to the other site, these demonstrate that off rate is slowed for both tracers when the unlabelled ligand is present.

-4- The authors indicated that they had mistakenly used a 'two binding sites' equation to fit the curves in Figure 4Bi in the first version and had replaced it with a 'one binding site'

equation. Nevertheless, two binding sites must be considered, one (site 2) being an allosteric binding site for site 1. No adjustment procedure is available in Prism to fit their data, but perhaps the authors can clarify that the 'one binding site' fitting does not fully describe reality.

-A statement outlining the choice of binding model and recognition that it does not fully reflect binding in this complex system has been added to the legend of Figure 4 (Line 876-879).

Minor points

- Suppl. Fig. 2: Y axis titles seem to be wrong: "% of untagged FFA1R maximum response" seem to be more accurate.

-Data are shown as the percent of response to T360 at untagged FFA1. This has been clarified in the y axis and legend for this figure.

- Line 407-412: "physical separation » again => « substantial distance" could be better.

-Corrected (lines 429-431)

Reviewer #4 (Remarks to the Author):

1. In Supplementary Figure 2A, TUG-905 is shown as a full agonist in the Ca²⁺ response (100% of T 360 response) with a pEC₅₀ close to 7.5 for wild-type receptors. This is inconsistent with other statements and data (Supplementary Table 1) in the manuscript regarding this ligand. The partial agonism appears only when the NLuc motif is coupled with the receptor (Supplementary Figure 2A). The impact of this behavior within the model needs to be analyzed.

-It is true that when using Flp-In cells expressing untagged FFA1 the efficacy of TUG-905 increases. However, while the efficacy of the partial agonist TUG-905 increases from the C terminal tag, to the N terminal tag, then is highest in the untagged, its potency does not change across these cell lines. In contrast, for the full agonist, T360, efficacy is unchanged across the different FFA1 constructs, but potency increases from the C terminal, to the N terminal tagged versions, then is highest at the untagged version. These data are entirely constant with the overall expression of FFA1 simply being highest in the untagged cell line, lower in the N terminal line, and lowest in the C terminal line. Because 905 is a partial agonist, the increased receptor expression manifests first in an increase in maximal response until it reaches the system maximum, whereas the full agonist is already at a system maximum so the expression reflects in a shift in potency. There is therefore no indication that the tags have substantially affected the ligand pharmacology at the receptor.

2. It is good that the manuscript mentions that the efficacies of TUG-2489 and TUG-

2490 cannot be fully determined (line 126, Figure 1B); however, this should also be noted in Supplementary Table 1.

-EMax values have been removed from the table and a note added that low potency did not allow for a clear determination of this parameter.

3. The statement “mean \pm SEM of pooled data from the indicated number of experiments” is still confusing. For pooled data, the value is not the mean but the best-fit value, and the standard error indicates the goodness of fit. This does not indicate the reproducibility of the experiment, which is usually required in this type of study.

-Mean \pm SEM here is referring to the individual data points on the graphs and curve fit values reported in tables for pEC50, pIC50m, pKi, EMax and % competition values. These parameters have been calculated independently from each experimental replicate, and the mean \pm SEM of these individual curve fit values is reported. Kinetic curve data is presented as the best fit value with 95% CI. A statement clarifying this point has been added to the methods section (line 638-641).

4. The calculation of kinetic parameters in Figures 5D and 5E is still confusing. The “one site exponential decay model” used here contains three parameters: starting point, ending point, and rate constant. All these parameters change and influence each other. Therefore, presenting only the rate constant does not fully describe the processes. Additionally, alternative multisite models may be more feasible in some of the cases studied and should be considered.

-We have added the non-specific binding parameter of the curve fits to Supplemental Table 4. We agree this is relevant, particularly for DISCo-BRET studies measuring off rate for TUG-2597, where some direct BRET from the C terminal NLuc to the ligand remains. We do not think there is value in including the Y0 values for the experiments however, as this value is simply a reflection of how much BRET was observed when we started measuring dissociation.

5. Figures 2F-H describe the BRET signal of TUG-2591, TUG-2355, and TUG-2287 binding to C and N terminal NLuc constructs. Analysis data have been provided only for the N terminal construct. Since the BRET signal of TUG-2591 for the C-terminal construct is almost at the same level as the BRET signal of TUG-2355 for the N-terminal construct, it should also be evaluated. The statement “very little specific BRET” on line 176 is not relevant here.

-Binding values (KD and BMax) with CIs have now been added to the legend of Figure 2. The level of BRET observed for these tracers to the C terminal NLuc construct it difficult to draw any conclusion from the affinity values obtained so these have not been discussed in the main manuscript. The wording of this sentence in the results has been modified to state that substantially less BRET was observed when using the C terminal construct than the N terminal construct for the same tracer ligand (Line 175-177).